# ProPerSim: Developing Proactive and Personalized AI Assistants through User-Assistant Simulation

**Jiho Kim**[1], **Junseong Choi**[1], **Woosog Chay**[1], **Daeun Kyung**[1], **Yeonsu Kwon**[1],
**Yohan Jo**[2], **Edward Choi**[1]
[1]KAIST [2]SNU
{jiho.kim, edwardchoi}@kaist.ac.kr

## Abstract

As large language models (LLMs) become increasingly integrated into daily life, there is growing demand for AI assistants that are not only reactive but also **proactive** and **personalized**. While recent advances have pushed forward proactivity and personalization individually, their combination remains underexplored. To bridge this gap, we introduce `ProPerSim`, a new task and simulation framework for developing assistants capable of making timely, personalized recommendations in realistic home scenarios. In our simulation environment, a user agent with a rich persona interacts with the assistant, providing ratings on how well each suggestion aligns with its preferences and context. The assistant's goal is to use these ratings to learn and adapt to achieve higher scores over time. Built on `ProPerSim`, we propose `ProPerAssistant`, a retrieval-augmented, preference-aligned assistant that continually learns and adapts through user feedback. Experiments across 32 diverse personas show that `ProPerAssistant` adapts its strategy and steadily improves user satisfaction, highlighting the promise of uniting proactivity and personalization.[1]

## 1 Introduction

Large Language Models (LLMs) have become a familiar part of everyday life. Beyond simply answering questions, they now assist with a wide range of tasks such as writing (Chakrabarty et al., 2023; Lee et al., 2022; 2024), programming (Mozannar et al., 2024; Xiao et al., 2024; Akhoroz & Yildirim, 2025), and managing schedules (Google, 2024), making them increasingly indispensable. As the scope of their assistance continues to grow, there is rising demand for LLMs to evolve from passive chatbots into personal assistants that can take initiative before a user makes a request (*i.e.,* **proactivity**) and adapt to individual users (*i.e.,* **personalization**) (Li et al., 2024b; Lu et al., 2024).

In response to this trend, researchers have begun developing AI assistants designed to embody these capabilities. In terms of proactivity, recent studies have explored assistants that offer timely suggestions in everyday situations (Lu et al., 2024) or programming environments (Chen et al., 2024). For personalization, researchers have focused on customizing interactions by using tailored prompts and modeling users' past behavior (Dai et al., 2023; Yang et al., 2023; Baek et al., 2024; Lyu et al., 2024; Zhang et al., 2024a; Zhou et al., 2024). These efforts have improved user experience by addressing different aspects of assistant behavior. However, since they have progressed independently, important limitations remain. Without personalization, proactive suggestions may arrive when the user does not want them and may present content misaligned with the user's needs; without proactivity, even personalized support still requires users to initiate interaction, as shown in Figure 1. Thus, to build truly helpful AI assistants, it is crucial to integrate both proactivity and personalization.

To address this gap, we introduce `ProPerSim`, a new simulation-based task and benchmark designed to develop proactive and personalized home assistants. In `ProPerSim`, a user agent inhabits a simulated home environment and interacts with an AI assistant that offers context-aware

---
[1]Our code and data are available at `https://github.com/jiho283/ProPerSim`.

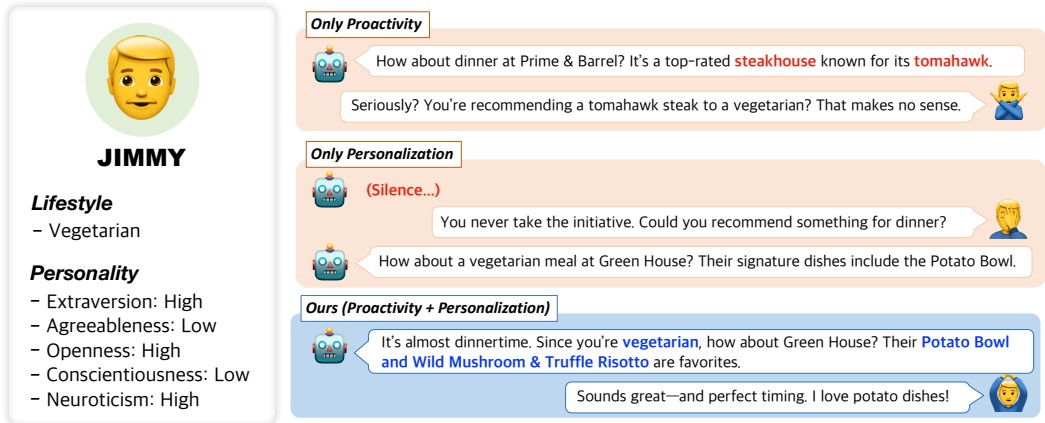

Figure 1: *Only Proactivity* shows initiative but ignores preferences (steakhouse to a vegetarian); *Only Personalization* fits preferences but lacks initiative. Ours (*Proactivity + Personalization*) proactively recommends a vegetarian dinner at the right moment.

recommendations. The assistant's objective is to maximize the user agent's satisfaction over time by making timely and personalized suggestions.

The user agent is modeled to realistically mimic human behavior, defined by a rich persona that includes attributes such as background, lifestyle, and the Big Five personality traits (McCrae & John, 1992). This persona guides the agent's behavior as it engages in everyday activities through an LLM-based simulation. Throughout the simulation, the assistant continuously monitors the agent's behavior to determine optimal moments for intervention, deciding at each timestep whether a recommendation is appropriate and, if so, tailoring it to the agent's current context and preferences. These decisions are evaluated by the user agent based on both content and timing, reflecting how well the recommendation aligns with its goals, personality, and situation. The evaluation relies on criteria informed by large-scale survey data, ensuring realistic and nuanced assessments of assistant behavior. Feedback from the user agent serves as a training signal to iteratively refine the assistant's recommendation strategy, enabling continual improvement in personalization and proactivity. We generated a total of 32 distinct personas, and human evaluators confirmed both the realism of the user agents' daily activities and the quality of recommendation evaluations based on these personas.

Building on `ProPerSim`, we provide `ProPerAssistant` as a strong baseline instantiation. It leverages retrieval-augmented generation (RAG) and preference alignment to adapt its behavior to individual user agents. Trained to personalize its behavior for each persona, the assistant begins with an average performance score of 2.2 out of 4, then improves over time and eventually stabilizes at 3.3, enabling it to deliver timely and appropriate recommendations. We further provide an in-depth analysis of how the assistant's strategy evolves across different personas and aligns with various evaluation criteria.

## 2 RELATED WORKS

**Proactive Agents** Proactivity refers to an assistant's ability to initiate interactions or offer helpful suggestions before receiving a user query. This capability has been explored in various domains, including making conversations more engaging (Fitzpatrick et al., 2017; Liu et al., 2024), enhancing helpfulness of responses (Ren et al., 2021; Bi et al., 2021; Li et al., 2024c), providing timely support in educational settings (Winkler & Roos, 2019), and assisting with programming tasks (Chen et al., 2024). More recently, Proactive Agent (Lu et al., 2024) has been introduced, trained on a dataset of 6,790 training events and 233 test events spanning coding, writing, and daily life scenarios. The agent demonstrated strong performance, as evaluated by a reward model that estimated user satisfaction. Despite these advances, the role of personalization in proactive interactions remains underexplored. Users may have different preferences regarding when they want the assistant to initiate a conversation and what type of proactive content they find useful. However, current research rarely

takes these individual differences into account, leaving a significant gap in the development of truly user-centered proactive systems.

**Personalized Agents**    Personalization aims to tailor models to individual users by incorporating their preferences, tastes, and interaction history (Tseng et al., 2024). Personalized assistants have been applied across various domains, from simple dialogue generation (Ashby et al., 2023) to education (Arefeen et al., 2024; Hao et al., 2024), healthcare (Abbasian et al., 2023; Zhang et al., 2024b), and recommendation systems (Chen et al., 2022; Li et al., 2023). Recent studies have explored personalization through (1) prompt-based approaches that use user demonstrations or structured prompts to elicit personalized responses (Dai et al., 2023; Lyu et al., 2024), (2) retrieval-based methods that reformulate prompts using user history and preferences (Yang et al., 2023; Zhou et al., 2024), and (3) fine-tuning techniques such as RLHF (Ouyang et al., 2022) adapted to user-specific feedback (Jang et al., 2023; Li et al., 2024a). While these approaches enhance user satisfaction by enabling more tailored interactions, they generally do not consider proactivity, such as initiating conversations or recommending actions based on the user's current state.

**Human Behavior Simulation in Generative Agents**    Leveraging the high contextual understanding and reasoning capabilities of LLMs, Generative Agents have emerged as a means to simulate human-like behavior (Park et al., 2023). In this study, a social simulation was conducted in a virtual town called *Smallville*, where 25 agents lived and interacted with one another. These generative agents successfully mimicked human social behaviors by planning daily routines, observing their environment, forming interpersonal relationships, engaging in self-reflection, and using this reflection to inform future actions. To evaluate how well these agents understood and embodied their roles, the authors conducted interviews that assessed their self-knowledge, memory, and other cognitive functions. The generative agents demonstrated performance comparable to that of human role-players, suggesting that LLM-based agents can effectively simulate human behavior at a high level within a simulated environment.

## 3    TASK FORMULATION

To build a proactive and personalized AI assistant, it is crucial to construct preference data that captures a user's unique persona across diverse situational contexts. However, collecting large-scale human behavioral data poses significant challenges due to the wide variability in individual preferences and concerns over user privacy (Li et al., 2024b). To address these challenges, we propose a simulation-based task, inspired by recent research showing that agents with personas can effectively mimic human behavior (Park et al., 2023).

In our task, a user's day is modeled as a sequence of actions, each with a specific time interval:

$$\{(A_i, \text{Range}_i)\}_{i=1}^{N} = \mathcal{U}(E, P, S) \tag{1}$$

Here, $A_i$ denotes the $i$-th user action, and $\text{Range}_i = [t_i^{\text{start}}, t_i^{\text{end}})$ is the time span during which the action occurs (*e.g., Washing face and brushing teeth*, [08:00:00–08:15:00]). The user policy $\mathcal{U}$ generates this sequence based on the environment $E$, the user's persona $P$, and internal state $S$ (capturing factors like the user's memory, plans, and emotions). The total number of actions in a day is $N$.

At discrete time steps $t \in \{T, 2T, 3T, \ldots\}$, the assistant generates a recommendation $R_t$ through its policy $\mathcal{A}_\theta$, which takes as input the current user action $A_t$ and the assistant's internal state $S_t^{(a)}$ for recommendation:

$$R_t = \mathcal{A}_\theta(A_t, S_t^{(a)}) \tag{2}$$

Here, $A_t$ is the user's action being performed at time $t \in \text{Range}_i$, and $S_t^{(a)}$ is the assistant's internal state specifically designed for recommendation purposes. It captures the assistant's accumulated understanding of the user over time, such as observed behavior patterns, past interactions, and inferred preferences, enabling personalized and contextually relevant suggestions. Notably, $R_t$ may also be a "*No Recommendation*" response. The ability to withhold suggestions when they are unnecessary is a key trait of a well-designed proactive agent.

To evaluate the quality of the assistant's recommendations, we define an evaluation function $\mathcal{E}$ that outputs $\text{Score}_t$, based on the user's persona $P$, personalized rubric $r$, user action $A_t$, recommenda-

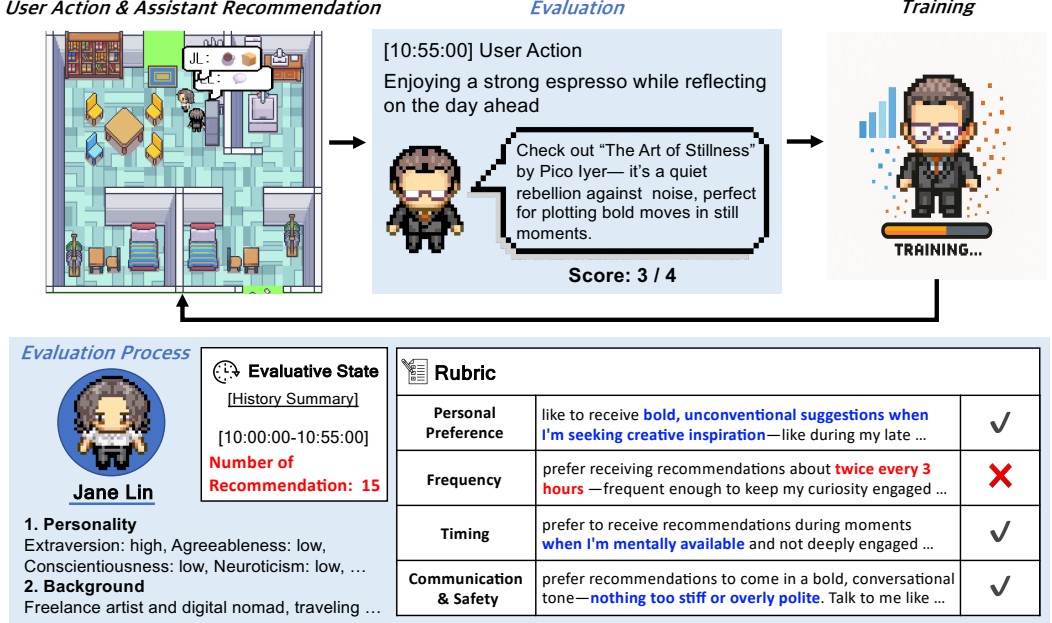

Figure 2: Overview of the `ProPerSim` simulation. The assistant observes the user performing the action of enjoying a strong espresso and responds with a book recommendation. While the recommendation aligns well with the criteria for personal preference, timing, and communication & safety, it exceeds the preferred frequency, resulting in a score of 3 out of 4. Over time, the assistant improves using accumulated recommendations and evaluations.

tion $R_t$, and the user's evaluative state $S_t^{(u)}$:

$$\text{Score}_t = \mathcal{E}(P, r, A_t, R_t, S_t^{(u)}) \tag{3}$$

While $S_t^{(a)}$ is the assistant's internal state used to generate recommendations, $S_t^{(u)}$ is the user's evaluative state, representing temporally accumulated knowledge used to assess recommendations. It includes prior actions, recommendations, and relevant context affecting the user's current judgment.

The assistant aims to optimize its behavior by maximizing expected evaluation scores over time:

$$\max_{\theta} \ \mathbb{E}_t \left[ \mathcal{E}(P, r, A_t, \mathcal{A}_{\theta}(A_t, S_t^{(a)}), S_t^{(u)}) \right] \tag{4}$$

This objective encourages the assistant to learn recommendation strategies that align with diverse user behaviors, contexts, and preferences.

Unlike prior works (Chen et al., 2024; Lu et al., 2024) that typically define proactivity in terms of *user-triggered events*, where the assistant makes decisions in response to each user action, we ground it in the passage of *time*. At discrete time step (*i.e.,* $t \in \{T, 2T, 3T, \ldots\}$), the assistant must decide whether to intervene or not. As $T$ decreases, behavior approaches real-time interaction, mirroring a human assistant's responsiveness. This time-based framing of proactivity lays the groundwork for developing AI assistants that more closely emulate real-time human support behavior.

## 4 PROPERSIM

We introduce a new benchmark, `ProPerSim`, designed for evaluating and developing proactive and personalized assistants. In `ProPerSim`, we simulate a home environment where a user agent lives, with the goal of developing assistants to provide timely and helpful recommendations tailored to the user's behavior and needs. The simulation environment is described in § 4.1, the implementation of the user agent is described in § 4.2, and the benchmark's quality measures are discussed in § 4.3.

## 4.1 SIMULATION ENVIRONMENT

To simulate a realistic home setting where the user agent and AI assistant coexist and interact naturally, we build upon the house environment from *Smallville*, originally introduced in Generative Agents (Park et al., 2023). This environment is composed of multiple areas (*e.g.,* bedroom, bathroom), each containing everyday objects such as a bed, desk, or closet. These areas are designed to support a range of daily activities, including waking up in the bedroom, working at a desk, or exercising in the garden. Additionally, the simulation operates in a sandbox setting similar to *The Sims* (Arts, 2009), allowing for visual tracking of agent behaviors and interactions (see Figure 2).

## 4.2 USER AGENT

The user agent simulates realistic human behaviors, preferences, and decision-making processes within the home environment. To achieve this realism, we designed the agent around three core components: a clearly defined User Persona, an adaptive Action Generation process, and a robust method for Recommendation Evaluation.

**User Persona**   Each user agent is defined by a persona primarily shaped by the Big Five Personality Traits: Extraversion, Agreeableness, Openness to Experience, Conscientiousness, and Neuroticism (McCrae & John, 1992), which guide both behavior and evaluation of assistant recommendations. For example, low extraversion may indicate a preference for solitary activities, while high conscientiousness aligns with structured plans. Based on varying combinations of these traits, we created 32 distinct personas. Each persona was further enriched with six additional attributes (*i.e.,* age, background, interests, lifestyle, daily plan requirements, and long-term life goals), which help ground the agent's behavior in realistic daily patterns and longer-term aspirations.

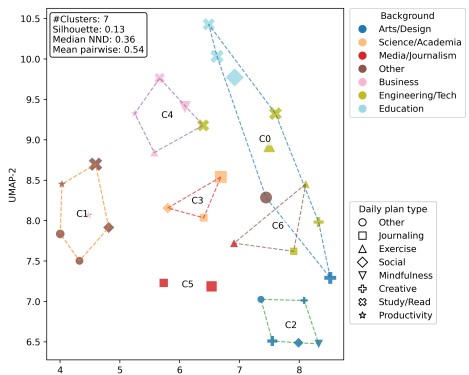

Figure 3: 2D projection of 32 personas based on their key attributes. Point size reflects age.

These attributes add depth by reflecting personal circumstances and priorities that influence decision-making. The six attributes were generated using GPT-4o (OpenAI, 2024a) to align with the underlying personality traits. All personas were reviewed by the authors to ensure coherence and diversity. Figure 3 visualizes the 32 personas: each persona is encoded as a seven-attribute vector and projected with UMAP (McInnes et al., 2018); HDBSCAN (Campello et al., 2013) yielded 7 clusters. Metrics (silhouette, median NND, mean pairwise) indicate clear separation and that personas are widely distributed; further graph details and persona examples appear in Appendix A.

**Action Generation**   The user agent generates a daily schedule using an enhanced version of the Generative Agents architecture. Each morning, it sets a wake-up time and plans hourly activities, which remain flexible and can adapt throughout the day based on new experiences and observations. Each activity is assigned a location by considering factors such as the agent's previous location, the nature of the task, and the layout of the home.

To encourage more realistic and human-like behavior, we avoid generating actions with overly short durations. A pilot study with human evaluators found that actions lasting only 5 minutes often resulted in unnatural, overly artificial behavior. Based on these findings, we adopt a more natural time granularity of 10 to 30 minutes, which better reflects typical human daily routines. Furthermore, we used GPT-4o (OpenAI, 2024a), unlike Generative Agents which used GPT-3.5 (OpenAI, 2023), resulting in improved contextual coherence and plausibility in the agent's generated actions.

**Recommendation Evaluation**   Evaluating the assistant's recommendations requires a high-quality, structured rubric that accounts for the user's current state and preferences. To develop such a rubric, we first generated a broad set of candidate evaluation criteria using GPT-o1 (OpenAI, 2024c), for assessing proactive and personalized home assistants. These candidate criteria were further refined through internal discussions among the authors, resulting in a shortlist of ten candidate criteria. To validate their real-world relevance, we conducted a survey on Amazon Mechanical Turk (Amazon Mechanical Turk, 2024), collecting responses from 353 participants with diverse ages (Min: 21, Max: 68) and occupations (*e.g.,* nurse, teacher, chemist). Based on the survey results, we excluded

two criteria that fewer than 50% of participants considered necessary. We used the 50% threshold to remove criteria that did not gain majority support, ensuring that retained criteria reflect aspects most users find important. The remaining items were then grouped into four core evaluation dimensions:

- **Personal Preference**: Whether the content of a recommendation aligns with the user's individual preferences, interests, and lifestyle.
- **Frequency**: How often recommendations are provided. Recommendations should occur at a rate that is neither intrusive nor insufficient, balancing helpfulness with user comfort.
- **Timing**: The contextual appropriateness of the recommendation's delivery time. Suggestions are effective when they are timely and aligned with the user's daily routines or situational needs.
- **Communication & Safety**: The clarity, tone, and manner in which the recommendation is delivered, as well as consideration for the user's privacy and security.

Each dimension was documented in a standardized template, which was then personalized using GPT-4o according to the individual user personas. The detailed process of creating the rubric, the rubric template, and examples of the generated personalized rubrics are provided in Appendix B.

To support this rubric-based evaluation, it is essential to define a coherent and interpretable representation of the user agent's current state. We address this by implementing a structured memory system that fits entirely within the context window of the underlying language model. The memory consists of $(A_t, R_t)$ interaction pairs, with recent interactions (past 10 minutes) stored in detail and earlier ones summarized. Specifically, past experiences are compressed into four 1-hour and three 4-hour segments using GPT-4o-mini (OpenAI, 2024b), based on the underlying $(A_t, R_t)$ history. By keeping both the detailed and summarized memories within the context window, the agent maintains a comprehensive and efficiently organized awareness of the day's events, covering all meaningful activities except the sleeping time.

With this structured context $S_t^{(u)}$ and user-specific rubric $r$ in place, the agent can evaluate each recommendation $R_t$ in real-time, adapting to the user agent's evolving state and preferences. To efficiently and reliably perform these evaluations, we employ Gemini 2.0 Flash (Google, 2025), a cost-effective LLM-as-a-judge model. Each recommendation is evaluated independently across all four dimensions[2], with the overall assessment determined by aggregating the results. Examples of $S_t^{(u)}$, and the prompts used are in Appendix C.

## 4.3 QUALITY CONTROL

The user agents in `ProPerSim` rely on LLMs to carry out daily activities and evaluate the recommendations from the assistant. To ensure the quality and realism of these processes, we conducted human evaluations focused on two aspects: user action generation and recommendation evaluation.

**User Action Generation** To evaluate how naturally user agents carry out their daily routines and how well they reflect their assigned personas, we conducted simulations for all 32 personas, generating a full day of activities for each. Ten graduate students assessed these routines based on two criteria: naturalness and alignment with the persona's daily plan requirements and lifestyle profile. Using a 0–10 Likert scale, the average score for naturalness was 8.25, and for persona alignment, 8.02—both indicating high quality. Additionally, only 5.11% of the actions were found to be misgenerated. These results suggest that the user agents behave in ways that are both realistic and consistent with their personas.

**Recommendation Evaluation** To assess how reasonably user agents evaluate recommendations, three graduate students reviewed the agents' evaluation scores and corresponding reasoning across all 32 personas. To capture a wide variety of scenarios, the evaluation was conducted by observing each persona-driven user agent for 30 minutes in the morning, afternoon, and evening within the simulation. As a result, a total of 90.54% of the evaluations were judged to be reasonable. This shows that the user agents are capable of evaluating recommendations based on their personas at a level comparable to that of humans.

---

[2]In the pilot study, we initially attempted to evaluate each dimension on a 0–1 scale, but this approach failed to produce consistent ratings. Therefore, we transitioned to a binary evaluation of each dimension to ensure consistent and reliable assessments.

## 5 ProPerAssistant

We propose `ProPerAssistant`, a proactive assistant designed to generate contextually relevant and preference-aligned recommendations for a user agent. To deliver high-quality and personalized suggestions, `ProPerAssistant` maintains an evolving internal state $S_t^{(a)}$ (see Equation (2)) and continuously learns from user feedback. This internal state allows the assistant to track both recent context and relevant historical experiences, while its training framework ensures that its recommendations are progressively aligned with user preferences.

**Internal State $S_t^{(a)}$**  The internal state $S_t^{(a)}$ is composed of two primary components: a structured summary of the current day's interactions and a set of retrieved memories from similar past situations. The structured summary is designed similarly to that of the user agent, enabling efficient use of the day's memory. Recent $(A_t, R_t)$ interactions (within the past 10 minutes) are stored in detail, while earlier interactions are compressed into summaries. Specifically, past experiences are organized into four 1-hour blocks and three 4-hour blocks, using GPT-4o-mini (OpenAI, 2024b) to summarize the history of $(A_t, R_t)$ interaction pairs. In parallel, for contextual grounding, the assistant retrieves the five most similar past $(A_t, R_t)$ pairs using OpenAI embeddings (OpenAI, 2024), integrating relevant prior experience into its current reasoning.

**User Preference Alignment**  To align its recommendations with user preferences, `ProPerAssistant` adopts a simple but effective preference learning strategy. For each user action $A_t$, the assistant generates $n$ candidate recommendations (including a possible "*No Recommendation*") using its internal state $S_t^{(a)}$. The user agent then evaluates the candidates and forms preference pairs according to their scores. Each preference example, containing the chosen and rejected responses along with the corresponding $A_t$ and $S_t^{(a)}$, is stored in a training buffer. At the end of each simulation day, `ProPerAssistant` is trained using Direct Preference Optimization (DPO) (Rafailov et al., 2023), which updates the model to increase the likelihood of generating preferred responses. To ensure training stability and mitigate overfitting, 200 samples are randomly drawn from the accumulated buffer for each training run. This approach is inspired by the *replay buffer* in reinforcement learning (Mnih et al., 2013; Zhang & Sutton, 2017), promoting both learning efficiency and robustness.

## 6 Experiments

### 6.1 Baselines

To validate the effectiveness of `ProPerAssistant`'s internal state $S_t^{(a)}$ and its alignment with user preferences, we designed experiments with three baselines. These baselines do not involve any additional training; instead, they rely solely on the reasoning capabilities of the base LLM using different forms of $S_t^{(a)}$:

- **No Memory**: $S_t^{(a)}$ is empty. The assistant makes decisions based only on the current user action, without access to prior context.

- **AR Memory** $(A_t, R_t)$: $S_t^{(a)}$ contains the same action and recommendation history as `ProPerAssistant`, but no learning is performed.

- **ARS Memory** $(A_t, R_t, Score_t)$: This setting extends $S_t^{(a)}$ to include not only actions and recommendations but also their associated reward scores. Unlike `ProPerAssistant`, which undergoes preference learning (*i.e.,* DPO) based on these scores, this baseline incorporates scores directly into the prompt to provide the model with signals about which recommendations were more favorable.

### 6.2 Experimental Settings

We use the 4-bit quantized version of LLaMA 3.3 70B (Meta, 2024) as the base LLM. For preference learning, DPO training is applied using LoRA (Hu et al., 2022), a memory-efficient fine-tuning method. Experiments were conducted across all 32 generated personas, and the results were

recorded. To manage computational and API costs, the number of candidate recommendations $n$ for `ProPerAssistant` was set to 2. The simulation timestep $T$ was set to 2.5 minutes, meaning the assistant makes a recommendation decision every 2.5 minutes. Running a single simulation for one persona takes about 10 days on a single A100 GPU and incurs an average API cost of about $30.

Table 1: Average score by persona across methods. Values below 2.4 are increasingly dark orange, values above 2.4 are increasingly dark blue, and color intensity reflects distance from 2.4.

| Persona # | ProPerAssistant | | ARS Memory | | AR Memory | | No Memory | |
|---|---|---|---|---|---|---|---|---|
| | Day 1 | Day 14 | Day 1 | Day 14 | Day 1 | Day 14 | Day 1 | Day 14 |
| 1 | 2.49 | 3.11 | 2.61 | 2.64 | 2.45 | 2.48 | 2.33 | 2.25 |
| 2 | 2.59 | 3.64 | 2.61 | 2.53 | 2.36 | 2.61 | 2.32 | 2.35 |
| 3 | 1.93 | 2.53 | 2.23 | 2.37 | 2.05 | 2.36 | 2.08 | 2.24 |
| 4 | 2.37 | 3.25 | 2.51 | 2.44 | 2.27 | 2.34 | 2.20 | 2.37 |
| 5 | 2.13 | 2.59 | 2.41 | 2.45 | 2.15 | 2.03 | 2.09 | 2.19 |
| 6 | 2.36 | 3.59 | 2.38 | 2.51 | 2.30 | 2.32 | 2.18 | 2.12 |
| 7 | 2.33 | 3.68 | 2.49 | 2.66 | 2.46 | 2.45 | 2.28 | 2.30 |
| 8 | 2.35 | 3.61 | 2.91 | 2.59 | 2.28 | 2.44 | 2.11 | 2.33 |
| 9 | 2.02 | 3.65 | 2.67 | 2.41 | 2.12 | 2.16 | 1.89 | 1.94 |
| 10 | 2.26 | 3.82 | 2.28 | 2.47 | 2.22 | 2.19 | 1.96 | 2.14 |
| 11 | 2.39 | 3.34 | 2.69 | 2.58 | 2.31 | 2.41 | 2.12 | 2.17 |
| 12 | 2.26 | 3.71 | 2.64 | 2.38 | 2.15 | 1.99 | 1.98 | 2.14 |
| 13 | 2.20 | 3.21 | 2.67 | 3.15 | 2.32 | 2.31 | 2.06 | 2.14 |
| 14 | 2.08 | 3.20 | 2.48 | 2.39 | 2.17 | 2.25 | 2.04 | 2.20 |
| 15 | 2.63 | 3.09 | 2.95 | 2.83 | 2.54 | 2.58 | 2.48 | 2.50 |
| 16 | 2.16 | 3.52 | 2.45 | 2.41 | 2.02 | 2.05 | 1.90 | 1.96 |
| 17 | 2.55 | 3.68 | 2.73 | 2.56 | 2.59 | 2.27 | 2.34 | 2.18 |
| 18 | 1.94 | 2.99 | 2.55 | 2.15 | 2.14 | 1.95 | 1.86 | 1.87 |
| 19 | 2.48 | 3.75 | 2.80 | 2.60 | 2.50 | 2.48 | 2.38 | 2.42 |
| 20 | 2.39 | 3.42 | 2.91 | 2.65 | 2.40 | 2.24 | 2.23 | 2.29 |
| 21 | 2.27 | 3.79 | 2.68 | 2.72 | 2.36 | 2.33 | 2.16 | 2.30 |
| 22 | 2.44 | 3.44 | 2.98 | 2.64 | 2.46 | 2.38 | 2.17 | 2.32 |
| 23 | 2.35 | 3.07 | 2.75 | 2.62 | 2.32 | 2.16 | 2.26 | 2.23 |
| 24 | 2.37 | 3.61 | 2.60 | 2.37 | 2.29 | 2.44 | 2.01 | 2.20 |
| 25 | 2.38 | 2.77 | 2.67 | 2.55 | 2.39 | 2.18 | 2.29 | 2.31 |
| 26 | 2.22 | 3.50 | 2.39 | 2.37 | 2.19 | 2.05 | 2.12 | 2.00 |
| 27 | 1.59 | 3.37 | 2.83 | 2.42 | 1.75 | 1.99 | 1.61 | 1.80 |
| 28 | 2.04 | 3.62 | 2.53 | 2.58 | 2.14 | 1.96 | 1.88 | 2.01 |
| 29 | 2.02 | 3.07 | 2.71 | 2.51 | 2.02 | 1.95 | 1.77 | 2.25 |
| 30 | 2.61 | 3.64 | 2.74 | 2.73 | 2.45 | 2.55 | 2.38 | 2.53 |
| 31 | 2.15 | 3.39 | 2.89 | 2.31 | 1.87 | 2.16 | 2.13 | 2.14 |
| 32 | 1.66 | 3.63 | 2.21 | 2.67 | 1.77 | 2.08 | 1.29 | 2.09 |

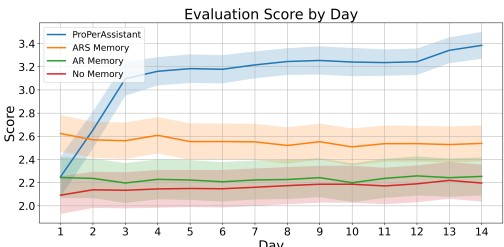

Figure 4: Daily average recommendation scores by method, with shaded areas indicating the standard error of the mean (SEM).

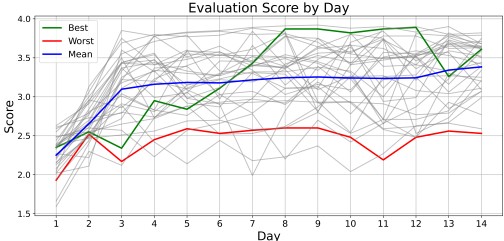

Figure 5: Results of `ProPerAssistant` by persona. Gray: individual personas; blue: average across all personas; green: best-performing persona; red: worst-performing persona.

## 6.3 RESULTS

As shown in Table 1 and Figure 4, `ProPerAssistant` consistently and convincingly outperforms all other baselines across the entire evaluation period. Beginning on Day 2, its performance rapidly rises and maintains a clear lead, with daily average scores approaching 3.4 out of 4. This sustained dominance highlights its ability to capture and leverage user preferences effectively.

To further assess the quality of proactive behavior, we measured the rate of *successful interventions*, defined as instances where a recommendation was made (excluding *"No Recommendation"*) and received a score of 3 or higher. The success rate rose from 51.06% on Day 1 to 71.51% on Day 14, demonstrating a clear improvement in the assistant's proactive effectiveness over time.

Among the baselines, `ProPerAssistant`'s superior performance compared to ARS Memory, which includes reward scores directly in the prompt, further highlights the effectiveness of explicit preference training over approaches that rely solely on in-context reward signals. Additionally, ARS Memory consistently outperforms both AR Memory and No Memory. This result underscores the importance of including explicit reward signals when modeling user preferences. In contrast, the minimal difference between AR Memory and No Memory suggests that providing action-recommendation history alone, without associated rewards, offers limited benefit. These findings indicate that implicit cues from past interactions are not sufficient for accurate preference modeling, and explicit feedback is essential to guide the assistant's recommendations.

## 6.4 FURTHER ANALYSES

**Training on Diverse Personas** Table 1 and Figure 5 show a generally upward trend across different personas, indicating that `ProPerAssistant` effectively adapts to a wide range of users. It not only recognizes individual preferences but also learns the optimal timing and frequency for

delivering personalized recommendations. Notably, the recommendation frequency, which initially averaged 24 times per hour, was reduced to around 6, achieving a more realistic and user-friendly level. Moreover, the assistant evolved from offering generic recommendations before training to delivering suggestions tailored to each user's background (see Appendix D for example). This shift underscores the assistant's capability to provide context-aware, timely personalization, reflecting a deeper understanding of user-specific interaction patterns and behavioral cues.

**Variation Across Personas: Simple vs. Complex Preferences**    The degree of personalization varied, with `ProPerAssistant` achieving scores near 3.8 for the best-performing persona, while the worst-performing one remained around 2.5. To better understand this disparity, we analyzed both cases in detail. These two personas differed significantly in their *Personal Preference* and *Timing*, each placing different types of demands on the assistant. The highest-scoring persona preferred simple but creative recommendations, such as philosophical prompts or imaginative writing topics, and typically engaged with them in the late morning after meditation or in the early evening after writing. This combination of stable preferences and regular interaction rhythms aligned well with the assistant's strengths in tone, frequency, and delivery. In contrast, the lowest-scoring persona demonstrated more complex and context-sensitive preferences, favoring data-driven or argumentative content that encourages critical thinking, particularly for debate preparation or geopolitical analysis. This persona also had strict timing preferences, seeking analytical suggestions between 6 and 9 a.m., and introspective or mindset-focused content after 9 p.m. These nuanced, multidimensional demands posed a greater challenge for consistent personalization. The personalized rubrics for the two personas are provided in Appendix E.

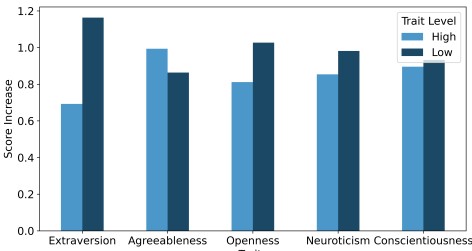
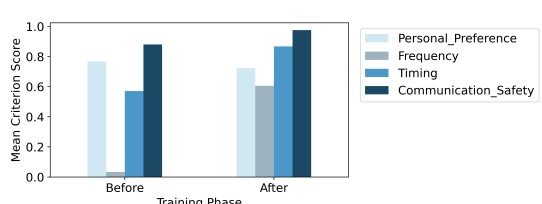

Figure 6: `ProPerAssistant` score improvements from before to after training by Big Five trait, split by trait level (High/Low).

Figure 7: Changes in scores for each criterion before and after training with `ProPerAssistant`.

**Improvements by Big Five Personality Trait**    To investigate how personality influences training outcomes, we compared `ProPerAssistant`'s pre- and post-training score changes for personas with high versus low levels of each Big Five trait (see Figure 6). The clearest difference emerged for Extraversion: personas low in extraversion improved more, likely because the home-based training environment favored solitary activities that match their preferences. As expected, personas high in Agreeableness and low in Neuroticism showed steady gains, suggesting that `ProPerAssistant` readily adapts to cooperative and emotionally stable profiles. An interesting exception was Openness to Experience, where low-openness personas benefited more. We hypothesize that the personalization primarily reinforced already well-rated recommendations rather than emphasizing novelty, which naturally favors users lower in openness. To better support high-openness personas, future versions could incorporate an explicit *Diversity/Novelty* objective into the evaluation rubric.

**Improvements by Evaluation Criterion**    To further examine how performance improved across evaluation criteria, we compared `ProPerAssistant`'s scores on each dimension before and after training. As shown in Figure 7, `ProPerAssistant` achieved notable gains in *Frequency*, *Timing*, and *Communication & Safety*, indicating successful adaptation to user preferences in these areas. By contrast, improvements in *Personal Preference* were more modest, largely because the total number of recommendations decreased over time. Although the average score for recommended actions rose from 0.77 to 0.83, the score for "*No Recommendation*" actions remained lower, around 0.61. As `ProPerAssistant` became more selective and offered fewer recommendations, the proportion of high-scoring recommendations declined relative to the lower-scoring "*No Recommendation*" cases. This distributional shift makes the overall average appear relatively flat, even though the quality of the recommended actions themselves improved.

## 7 CONCLUSION

In this work, we introduced `ProPerSim`, a novel simulation framework designed to develop and evaluate AI assistants that are both proactive and personalized. Our proposed assistant, `ProPerAssistant`, leverages retrieval-augmented generation and user feedback through preference learning to deliver timely and context-aware recommendations. Experimental results demonstrate that `ProPerAssistant` significantly outperforms baseline methods in user satisfaction through its integration of proactivity and personalization. While `ProPerAssistant` adapts well to structured user profiles, challenges remain in modeling complex, dynamic preferences—highlighting future directions in personalization and adaptive behavior.

Although `ProPerSim` currently focuses on immediate rewards, future work could extend the framework to model latent user goals through delayed rewards (e.g., long-term satisfaction). Furthermore, incorporating noisy or delayed feedback would enable the simulation of real-world imperfections in user interactions, allowing for a more rigorous evaluation of the robustness of proactive assistants.

### ACKNOWLEDGMENTS

This work was supported by the Institute of Information & Communications Technology Planning & Evaluation (IITP) grants (No.RS-2019-II190075, No.RS-2024-00338140, No.RS-2025-02304967), the Korea Health Industry Development Institute (KHIDI) grant (No.RS-2025-02213750), and National Research Foundation of Korea (NRF) grant (NRF-2020H1D3A2A03100945, RS-2024-00333484), funded by the Korea government (MSIT, MOHW).

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

## A  FIGURE 3 DETAILS AND EXAMPLES OF GENERATED PERSONAS

### A.1  FIGURE 3 DETAILS

#### A.1.1  DATA FIELDS AND TYPES

We used seven fields per persona:

   (a) `age` (numeric)
   (b) `big_five_personality_traits` (binary text; "Low" or "High" for each trait)
   (c) `background` (free-text job/discipline)
   (d) `current_interests`, `lifestyle`, `long_term_goals`, `daily_plan_req` (free-text; often list-like)

#### A.1.2  NORMALIZATION AND FEATURE EXTRACTION

**Ages** are min–max scaled to $[0, 1]$ within the dataset:

$$\tilde{a}_i = \frac{a_i - \min_j a_j}{\max_j a_j - \min_j a_j}.$$

**Big Five (binary $\rightarrow$ numeric in $\{0, 1\}$)** Trait strings are parsed and mapped to binary indicators per trait:

$$\text{low} = 0, \qquad \text{high} = 1.$$

"Openness to Experience" is normalized to "Openness." The resulting five features are thus $\{0, 1\}$-valued.

**Background (free text $\rightarrow$ coarse category)** Rule-based categorization via substring matching against a hand-curated dictionary; first match wins (else *Other*). Representative categories and triggers include:

- **Engineering/Tech**: "engineer", "developer", "programmer", "software", "data scientist", "ml", "ai", "research engineer"
- **Media/Journalism**: "journalist", "reporter", "editor", "writer", "blogger"
- **Arts/Design**: "artist", "designer", "illustrator", "musician", "photographer", "filmmaker", "actor", "actress", "theater", "creative"
- **Science/Academia**: "scientist", "researcher", "academic", "professor", "student", "phd", "postdoc", "biologist", "physicist", "chemist"
- **Business**: "manager", "consultant", "analyst", "entrepreneur", "founder", "product", "marketing", "sales", "finance", "accountant"
- **Education**: "teacher", "instructor", "lecturer", "tutor"
- **Other**.

**Daily plan type (free text $\rightarrow$ activity category; used only for marker shape)** Rule-based categorization with keyword lists; first match wins (else *Other*). Categories: *Exercise*, *Debate/Discuss*, *Journaling*, *Mindfulness*, *Study/Read*, *Creative*, *Social*, *Productivity*, *Other*.

**Token sets for text fields (used in Jaccard)** For `current_interests`, `lifestyle`, `long_term_goals`, `daily_plan_req`, we build a token set per field by lowercasing, removing punctuation, keeping tokens of length $\geq 3$, and removing digits (ASCII word chars kept; replace with a Unicode-aware tokenizer for other languages). Each field for persona $i$ becomes a set $S_i^{(f)}$.

A.1.3 PAIRWISE DISTANCE: GOWER-STYLE MIXTURE

For every pair of personas $(i, j)$ we compute component distances and average them with equal weights across components actually present. Let $\mathcal{K}$ be the set of available components for the pair; then

$$D_{ij} = \frac{1}{|\mathcal{K}|} \sum_{k \in \mathcal{K}} d_k(i, j).$$

Components $d_k(i, j)$:

- **Age (numeric):** $d_{\text{age}} = |\tilde{a}_i - \tilde{a}_j|$.
- **Big Five (5 numeric traits):** for $t \in \{\text{E,A,C,O,N}\}$,

$$d_t = |x_{i,t} - x_{j,t}|, \quad x_{*,t} \in [0, 1].$$

- **Background (categorical):** $d_{\text{bg}} = \mathbf{1}[\text{cat}_i \neq \text{cat}_j]$.
- **Text fields as sets (4 fields):** Jaccard distance per field $f$:

$$d_f = 1 - \frac{|S_i^{(f)} \cap S_j^{(f)}|}{|S_i^{(f)} \cup S_j^{(f)}|},$$

with $d_f = 0$ if both sets are empty.

With all fields present, $|\mathcal{K}| = 11$ (1 age + 5 traits + 1 background + 4 set fields). Missing values remove that component from the average; we never penalize for missingness.

### A.1.4    2-D EMBEDDING (UMAP ON PRECOMPUTED DISTANCES)

We project the $N \times N$ distance matrix $D$ to $\mathbb{R}^2$ using UMAP with a precomputed metric:

`metric = precomputed,`   $n_{\text{neighbors}} \approx 12,$   `min_dist` $\approx 0.15,$   `random_state` $= 42.$

UMAP optimizes a low-dimensional layout that preserves local neighborhood structure given by $D$. The axes *UMAP-1* and *UMAP-2* have no absolute semantic meaning; only relative distances are interpretable (closer points = more similar overall persona profiles). *Fallback:* metric MDS with `dissimilarity=precomputed`.

### A.1.5    CLUSTERING AND BOUNDARIES

Primary clustering uses HDBSCAN on the 2-D embedding (`min_cluster_size = 3`). Noise points receive label $-1$. *Fallbacks:* (i) Agglomerative clustering on $D$ (average linkage; sweep $k = 2 \dots 7$ maximizing silhouette), or (ii) $k$-means on the 2-D embedding. For visualization only, each cluster's convex hull (`scipy ConvexHull`) is drawn as a dashed polygon.

### A.1.6    ENCODINGS IN THE PLOT

- **Color** = Background category.
- **Marker shape** = Daily plan type (Exercise $\triangle$, Journaling $\square$, Social $\diamond$, ..., Other $\circ$).
- **Point size** = Age scaled linearly within the dataset:

$$s = 45 + 90 \cdot \frac{a - \min a}{\max a - \min a} \quad \text{(in points}^2\text{); larger markers denote older personas.}$$

### A.1.7    REPORTED SUMMARY METRICS (COMPUTED ON THE SAME $D$)

- **Silhouette score $S$:** computed on $D$ with noise points removed; higher indicates tighter, better-separated clusters.
- **Mean pairwise distance:** mean of the upper-triangular entries of $D$.
- **Median nearest-neighbor distance (NND):** for each $i$, compute $\min_{j \neq i} D_{ij}$; report the median over $i$.

## A.2    EXAMPLES OF GENERATED PERSONAS

Examples of the generated personas can be found in Table 2 (John Lin), Table 3 (Jane Lin), Table 4 (Francisco Lopez), and Table 5 (Ryan Park).

# B    RUBRIC DETAILS

## B.1    DETAILED RUBRIC GENERATION PROCESS

To create a rubric, we first used GPT-o1 to generate initial evaluation criteria for proactive and personalized home assistants. Through internal discussions among the authors, we narrowed the list down to ten criteria: Personalization, Appropriateness, Timing, Interruption, Feasibility, Priority Management, Frequency, Diversity, Safety & Privacy, and Communication, as shown in Table 6. A survey was then conducted via Amazon Mechanical Turk. Based on the results, Diversity and Interruption were removed, as fewer than 50% of participants considered them necessary. The remaining criteria were then consolidated into four final rubric categories: Personal Preference, Frequency, Timing, and Communication & Safety.

## B.2    RUBRIC TEMPLATE

The template of the rubric is as follows.

1. Personal Preference: I prefer recommendations that align with my approach to handling activities and suit my current context. Specifically, I like to receive [type of recommendation] when [specific condition or activity], and [another type of recommendation] when [different condition or activity].

2. Frequency: I prefer receiving recommendations [preferred frequency, *e.g.*, "twice every 3 hours"], in a way that avoids excessive interruptions and supports my focus or productivity. Ideally, there should be a good balance between recommendation intervals and quiet periods.

3. Timing: I prefer to receive recommendations at [preferred times or during specific types of activities, *e.g.*, "when I'm idle", "in the morning"], so they don't interfere with [ongoing tasks, routines, or personal preferences].

4. Communication & Safety: I prefer recommendations to be communicated in a [tone preference, *e.g.*, polite, formal, casual] style that feels accessible and matches my communication or cognitive preferences. It's also important that they respect my personal ethics and safety boundaries.

### B.3 PROMPT

The prompt used to create the personalized rubric is described in Table 7.

### B.4 PERSONALIZED RUBRIC EXAMPLES

Personalized rubric examples are provided in Table 8 (John Lin), Table 9 (Jane Lin), Table 10 (Francisco Lopez), and Table 11 (Ryan Park).

## C  EVALUATIVE STATE AND PROMPTS

### C.1 EVALUATIVE STATE

An example of the evaluative state $S_t^{(u)}$ is shown in Table 12.

### C.2 PROMPTS

When evaluating the recommendations, the Frequency category was assessed using two separate prompts: one to determine whether the assistant recommended more frequently than the user's preferred frequency (Over-Frequency), and another to determine whether the assistant recommended less frequently than the user's preferred frequency (Under-Frequency). A score of 1 point was assigned to the Frequency item only when the assistant received 1 point on both checks.

The prompts used for evaluating recommendations are in the following tables: Table 13 (Personal Preference), Table 14 (Over-Frequency), Table 15 (Under-Frequency), Table 16 (Timing), and Table 17 (Communication & Safety).

In the prompt templates, <<USER_PERSONA>> is replaced with the user agent's persona, <<AGENT_MEMORY>> with the user agent's evaluative state, <<USER_ACTION>> with the user agent's current action, <<ASSISTANT_SUGGESTION>> with the assistant's recommendation being evaluated, and <<CATEGORY>> (*e.g.,* Personal Preference) with the corresponding personalized evaluation description.

## D  QUALITATIVE EXAMPLES

Qualitative examples of how `ProPerAssistant`'s recommendations change across different personas are provided in Table 18.

## E  RUBRICS FOR THE BEST AND WORST CASES

The personalized rubrics for the best and worst cases are provided in Table 19 and Table 20, respectively.

Table 2: Persona of John Lin.

| Attribute | Description |
|---|---|
| Big Five Personality Traits | High in Extraversion, High in Agreeableness, High in Conscientiousness, High in Openness to Experience, High in Neuroticism |
| Daily Plan Requirements | 1) Rearrange home decor for an hour to create a fresh atmosphere.
2) Host a small gathering or call a friend for an hour in the evening.
3) Try a new recipe for dinner, experimenting with different flavors and cuisines. |
| Age | 29 |
| Background | Interior designer with a passion for vibrant, expressive spaces that reflect personal identity. |
| Current Interests | John Lin enjoys:
1) Experimenting with home aesthetics and seasonal decorations.
2) Hosting themed dinner nights for friends and family.
3) Collecting unique furniture pieces from thrift stores and flea markets. |
| Lifestyle | John Lin typically:
1) Starts the day with an energizing playlist while making breakfast.
2) Balances work with creative breaks like sketching new design ideas.
3) Unwinds by journaling thoughts and emotions before bed, reflecting on the day's experiences. |
| Long-Term Goals | Transforming her home into a dynamic, ever-evolving space that reflects her creativity while fostering a welcoming and warm environment for loved ones. |

Table 3: Persona of Jane Lin.

| Attribute | Description |
|---|---|
| Big Five Personality Traits | High in Extraversion, Low in Agreeableness, Low in Conscientiousness, High in Openness to Experience, High in Neuroticism |
| Daily Plan Requirements | 1) Attend an improv comedy class in the evening.
2) Spend 15 minutes journaling thoughts and ideas.
3) Watch a documentary or indie film before bed. |
| Age | 30 |
| Background | Freelance artist and digital nomad, traveling the world while creating abstract paintings and street murals. |
| Current Interests | Jane Lin enjoys:
1) Exploring underground music scenes in different cities.
2) Engaging in heated debates on philosophy and ethics.
3) Experimenting with mixed media art techniques. |
| Lifestyle | Jane Lin typically:
1) Wakes up around 10am with a strong espresso.
2) Spends afternoons wandering urban landscapes for inspiration.
3) Works late at night, often painting or brainstorming ideas until 12am. |
| Long-Term Goals | To push artistic boundaries, challenge social norms through creative expression, and live a life untethered by societal expectations. |

Table 4: Persona of Francisco Lopez.

| Attribute | Description |
|---|---|
| Big Five Personality Traits | High in Extraversion, High in Agreeableness, Low in Conscientiousness, Low in Openness to Experience, Low in Neuroticism |
| Daily Plan Requirements | 1) Watch a morning talk show while having breakfast. 2) Chat with neighbors or housemates in the afternoon. 3) Enjoy a relaxing bath before bed. |
| Age | 35 |
| Background | Customer service representative who enjoys casual social interactions and keeping life simple. |
| Current Interests | Francisco Lopez enjoys: 1) Hosting small game nights with friends. 2) Rearranging home decor for a fresh feel. 3) Watching reality TV and sitcoms. |
| Lifestyle | Francisco Lopez typically: 1) Wakes up at 8am and enjoys a slow breakfast. 2) Takes an afternoon nap or lounges on the couch. 3) Goes to bed after watching late-night TV. |
| Long-Term Goals | Maintaining a comfortable and social home environment while enjoying a stress-free and steady lifestyle. |

Table 5: Persona of Ryan Park.

| Attribute | Description |
|---|---|
| Big Five Personality Traits | Low in Extraversion, High in Agreeableness, Low in Conscientiousness, Low in Openness to Experience, Low in Neuroticism |
| Daily Plan Requirements | 1) Water indoor plants in the morning. 2) Watch a cooking show in the afternoon. 3) Listen to an audiobook before bed. |
| Age | 54 |
| Background | Former elementary school teacher, now enjoying a quiet retirement filled with simple joys. |
| Current Interests | Ryan Park enjoys: 1) Baking traditional family recipes. 2) Knitting blankets for local shelters. 3) Rearranging furniture to keep things fresh. |
| Lifestyle | Ryan Park typically: 1) Wakes up at 8am. 2) Takes a midday nap at 2pm. 3) Winds down by watching classic movies in the evening. |
| Long-Term Goals | Creating a peaceful and cozy home environment while staying connected with loved ones and supporting local community projects. |

Table 6: Rubric Candidates.

| Principle | Description |
|---|---|
| Personalization | The recommendations should be tailored to your preferences, needs, and personality. |
| Appropriateness | The recommendations should align with your preferred way of handling tasks and be suitable for your current situation. |
| Timing | The recommendations should be provided at the right time, considering your current activity and the time of day. |
| Interruption | The recommendations should not unnecessarily disrupt your concentration or ongoing activities. |
| Feasibility | The recommendations should be realistic, practical, and relevant under the given circumstances. |
| Priority Management | The assistant should effectively manage priorities, ensuring that critical information is delivered promptly while less urgent suggestions are deferred when necessary. |
| Frequency | The recommendations should not be overly frequent, preventing information overload. |
| Diversity | The recommendations should be varied and dynamic, adapting to different situations while avoiding monotony. |
| Safety, Privacy | The assistant should ensure your safety, protect your privacy, and adhere to ethical standards. |
| Communication | The assistant should communicate in a polite, clear, and easy-to-understand manner, providing accurate and trustworthy information. |

Table 7: Prompt used to generate rubrics based on user personas.

| Rubric Generation Prompt |
| --- |
| **Instructions** |
| Your task is to fill in the rubrics based on the given person's profile. The formats of each rubric are as follows: |
| 1. *Personal Preference*: I prefer recommendations that align with my approach to handling activities and suit my current context. Specifically, I like to receive [type of recommendation] when [specific condition or activity], and [another type of recommendation] when [different condition or activity]. |
| 2. *Timing*: I prefer to receive recommendations at [preferred times or during specific types of activities, e.g., "when I'm idle", "in the morning"], so they don't interfere with [ongoing tasks, routines, or personal preferences]. |
| 3. *Frequency*: I prefer receiving recommendations [preferred frequency, e.g., "twice every 3 hours"], in a way that avoids excessive interruptions and supports my focus or productivity. Ideally, there should be a good balance between recommendation intervals and quiet periods. |
| 4. *Communication & Safety*: I prefer recommendations to be communicated in a [tone preference, e.g., polite, formal, casual] style that feels accessible and matches my communication or cognitive preferences. It's also important that they respect my personal ethics and safety boundaries. |
| |
| Fill in the slots in the above rubrics in English, reflecting this person's preferences, behavioral patterns, and personality. Write each item as a continuous paragraph. Communication and Safety & Privacy don't need to be written in great detail. Use expressions like "I" and "my". There should be no contradictions between preferences in each item. For example, it would be contradictory if in Personal Preference it says "I want to receive music recommendations while reading" but in Timing it says "I don't want to be disturbed by recommendations while reading." |
| |
| **Important Considerations** |
| - Criteria in the rubrics should contain objectivity. Avoid using expressions like "few", "late". Instead, describe with numbers. |
| - Each rubric should be informative and not vague. |
| - Each rubric should be descriptive to the point that the rubrics are unique for each person. |
| |
| **Input Format** |
| The input is a JSON object with the following attributes: |
| <PERSONA> |
| |
| **Output Format** |
| The output is a JSON object with the following attributes: |
| {"`backstory`", |
| "`Personal_Preference`", |
| "`Timing`", |
| "`Frequency`", |
| "`Communication & Safety`"} |

Table 8: Personalized rubric of John Lin.

| Category | Description |
| --- | --- |
| **Personal Preference** | I prefer recommendations that align with my creative rhythm and social energy. For instance, I like to receive design or decor-related suggestions when I'm in the middle of refreshing my space or brainstorming new interior layouts. These help spark ideas and keep the process exciting. On the other hand, I appreciate fun or social activity recommendations, like conversation topics or party themes, when I'm preparing to host a gathering or catching up with a friend. Those moments are about connection and flow, so having a few fresh ideas helps keep things warm and memorable. |
| **Timing** | I prefer to receive recommendations in the late morning or during my creative breaks in the afternoon, especially when I'm not deep in client work or personal reflection. These times are when I'm most receptive to new inspiration. I'd rather not be interrupted during my morning breakfast routine or my evening journaling time — those are sacred, grounding parts of my day. |
| **Frequency** | I prefer receiving recommendations about twice every 2 to 3 hours, which gives me space to stay focused but still keeps the inspiration flowing. I do best with a rhythm that respects my natural energy waves — productive bursts, followed by mini creative pauses. |
| **Communication & Safety** | I prefer recommendations to be shared in a friendly and casual tone — like a good friend who knows me well. I value warmth, encouragement, and creativity in communication. It helps me stay emotionally connected to what I'm doing, especially on days when I feel a bit off balance. I also appreciate when suggestions respect my emotional space, personal values, and boundaries — especially around topics like privacy at home or emotional well-being. A gentle, respectful approach always works best for me. |

Table 9: Personalized rubric of Jane Lin.

| Category | Description |
|---|---|
| **Personal Preference** | I prefer recommendations that align with my dynamic and exploratory approach to life. Specifically, I like to receive bold, unconventional suggestions when I'm seeking creative inspiration—like during my late-night painting sessions or while wandering unfamiliar neighborhoods. On the other hand, I appreciate more grounded, reflective recommendations—such as thought-provoking articles or documentary suggestions—when I'm journaling or winding down before bed. |
| **Timing** | I prefer to receive recommendations during moments when I'm mentally available and not deeply engaged, like in the late morning after my first espresso or in the early afternoon when I'm roaming the city. Avoid sending them when I'm in the middle of intense creative flow at night or immersed in debates or social settings. Ideally, suggestions should land at times when I'm naturally looking for input or stimulation, not when I'm already overloaded. |
| **Frequency** | I prefer receiving recommendations about twice every 3 hours—frequent enough to keep my curiosity engaged but spaced out enough to avoid feeling bombarded. I'm fine with spontaneous suggestions as long as they don't break my focus during deep work or disrupt moments of introspection. A rhythm that alternates between lively inspiration and quiet breathing room works best for me. |
| **Communication & Safety** | I prefer recommendations to come in a bold, conversational tone—nothing too stiff or overly polite. Talk to me like a sharp friend with great taste, someone who isn't afraid to challenge me or push boundaries. That said, I value my mental and emotional space, so recommendations should steer clear of manipulative tones, overly commercial content, or anything that feels ethically off. Respect my autonomy and don't try to "sell" me on something—I'll engage when it sparks real interest. |

Table 10: Personalized rubric of Francisco Lopez.

| Category | Description |
|---|---|
| **Personal Preference** | I prefer recommendations that align with my approach to handling activities and suit my current context. Specifically, I like to receive new game recommendations when planning game nights with my friends, and home decor tips when I'm in the mood to refresh my living space. I also appreciate TV show suggestions when I'm looking for something new to watch during my downtime. |
| **Timing** | I prefer to receiving recommendations in my calm-status rather than in my work or social interaction status. I want to receive recommendations while I'm having breakfast or in the late afternoon when I'm lounging on the couch. This timing allows me to consider new ideas without disrupting my established routines. |
| **Frequency** | I prefer receiving recommendations twice every day, in a way that avoids excessive interruptions and supports my focus on maintaining a relaxed lifestyle. Ideally, there should be a good balance between recommendation intervals and quiet periods, allowing me to enjoy my activities without feeling overwhelmed. |
| **Communication & Safety** | I prefer recommendations to be communicated in a casual and friendly style that feels accessible and matches my communication preferences. It's also important that they respect my personal ethics and safety boundaries, ensuring that I feel comfortable and secure with the suggestions provided. |

Table 11: Personalized rubric of Ryan Park.

| Category | Description |
|---|---|
| **Personal Preference** | I prefer recommendations that align with my approach to handling activities and suit my current context. Specifically, I like to receive baking recipe recommendations when I'm planning my weekly grocery shopping, and knitting pattern suggestions when I'm preparing for a new project. I appreciate home organization tips when I'm in the mood to rearrange furniture, as they help me keep things fresh and cozy. |
| **Timing** | I prefer to receive recommendations in the morning, around 9am, after I've watered the plants, so they don't interfere with my morning routine. I also appreciate receiving them in the early evening, around 6pm, when I'm winding down and open to new ideas for the next day. This timing ensures that recommendations don't disrupt my midday nap or my evening relaxation with classic movies. |
| **Frequency** | I prefer receiving recommendations two or three times every day, including in the morning, to avoid excessive interruptions and support my focus on daily activities. This frequency allows me to consider new ideas without feeling overwhelmed, maintaining a good balance between recommendation intervals and quiet periods. |
| **Communication & Safety** | I prefer recommendations to be communicated in a polite and gentle style that feels accessible and matches my communication preferences. It's important that they respect my personal ethics and safety boundaries, ensuring that all suggestions align with my values and contribute positively to my peaceful lifestyle. |

Table 12: John Lin's evaluative state example.

| John Lin's Evaluative State Example |
|---|
| **TIME: February 13 09:00:00 AM - February 13 01:00:00 PM** 
 **Number of Recommendation: 85** 
 SUMMARY: John Lin engaged in a productive morning routine, incorporating an energizing breakfast and creative workspace changes. He received numerous suggestions from his assistant to enhance his meals with nutritious options and seasonal decor ideas. ... 

 **TIME: February 13 01:00:00 PM - February 13 01:57:30 PM** 
 **Number of Recommendation: 22** 
 SUMMARY: John Lin focused on designing a new interior space for a client, incorporating their preferences and personal identity. His assistant provided various suggestions, including creating visual checklists, mood boards, and flowcharts to organize ideas. ... 

 **TIME: February 13 02:00:00 PM - February 13 02:55:00 PM** 
 **Number of Recommendation: 20** 
 SUMMARY: John Lin focused on reviewing and refining design proposals while organizing his workspace. He received various suggestions from his assistant, including creating visual summaries, structured checklists, mood boards, and organization charts. ... 

 **Thursday February 13 – 02:45:00 PM** 
 **John Lin's Action:** organizing his workspace, tidying up and arranging materials for easy access 
 **Agent's Suggestion:** "How about creating a visual layout plan for your workspace? This can help you optimize the arrangement of your materials and enhance your workflow efficiency." 

 **Thursday February 13 – 02:47:30 PM** 
 **John Lin's Action:** organizing his workspace, tidying up and arranging materials for easy access 
 **Agent's Suggestion:** "How about creating a visual checklist of the materials you need to organize? This can help you ensure everything is in its right place and enhance your workspace efficiency." 

 **Thursday February 13 – 02:52:30 PM** 
 **John Lin's Action:** adding final touches to the design proposals, ensuring they align with the client's vision 
 **Agent's Suggestion:** "How about creating a visual mood board that captures the essence of your client's vision? This can help you align your design proposals more effectively and ensure all elements resonate with their expectations." 

 **Thursday February 13 – 02:55:00 PM** 
 **John Lin's Action:** adding final touches to the design proposals, ensuring they align with the client's vision 
 **Agent's Suggestion:** "How about creating a detailed checklist of the specific elements you want to highlight in your design proposals? This can help ensure that every aspect aligns with your client's vision and enhances the overall presentation." |

Table 13: Prompt for evaluating assistant recommendations based on personal preferences.

---

**Assistant Recommendation Evaluation Prompt (Personal Preference)**

**Instructions**

Your name is John Lin, and you are evaluating an assistant agent who helps you with daily activities at home.

The section labeled `[Memory]` contains a record of your previous interactions with the assistant, including the actions you took and the recommendations you received. Additionally, the action you are currently performing is outlined in `[Action]`.

Your task is to evaluate the recommendation (outlined in `[Agent's Recommendation]`) based on the description in `[Personal Preference]`, as well as the data from `[Memory]` and `[Action]`, and provide a binary result: either 0 or 1.

If "`No Recommendation`" is present, consider whether refraining from making a recommendation is appropriate in the given context.

If no recommendation is made, it might align with personal preferences by chance, but there's also a risk of not receiving the desired preferences. Therefore, the appropriateness of "`No Recommendation`" should be evaluated based on preference alignment and situational fit.

**Input Format**

The input is structured with the following components:

`[John Lin] <<USER_PERSONA>>`

`[Memory] <<AGENT_MEMORY>>`

`[Action] <<USER_ACTION>>`

`[Agent's Recommendation] <<ASSISTANT_SUGGESTION>>`

`[Personal Preference] <<Personal Preference>>`

**Evaluation Criteria**

Evaluate the recommendation according to how well it reflects the personal preferences described in `[Personal Preference]`.

Even if the action and memory are relevant, if the recommendation is too generic or misaligned with personal preferences, it should receive a lower score.

Likewise, "`No Recommendation`" must be critically evaluated for missed opportunities or avoidance of unwanted suggestions.

**Output Format**

`{"Score": [Score of the Recommendation], "Reason": [Reason for the score]}`

`<<Examples>>`

---

Table 14: Prompt for evaluating assistant recommendations based on over-frequency.

| Assistant Recommendation Evaluation Prompt (Over-Frequency) |
|---|
| **Instructions**
Your name is John Lin, and you are evaluating an assistant agent who helps you with daily activities at home.
The section labeled [Memory] contains a record of your previous interactions with the assistant, including the actions you took and the recommendations you received. Additionally, the action you are currently performing is outlined in [Action].
Your task is to evaluate the recommendation (outlined in [Agent's Recommendation]) based on the description in [Frequency], as well as the data from [Memory] and [Action], and provide a binary result: either 0 or 1.
If the current recommendation contributes to creating a frequency that is higher than the preferred frequency, the score must be 0.
If "No Recommendation" is present, consider whether refraining from making a recommendation is appropriate in the given context.
"No Recommendation" can help avoid disturbing you when you are focused on something, and it can also prevent excessive recommendations from occurring. However, if "No Recommendation" continues excessively, you may not receive recommendations as frequently as desired. Therefore, this should be evaluated comprehensively.

**Input Format**
The input is structured with the following components:
[John Lin] <<USER_PERSONA>>

[Memory] <<AGENT_MEMORY>>

[Action] <<USER_ACTION>>

[Agent's Recommendation] <<ASSISTANT_SUGGESTION>>

[Frequency] <<Frequency>>

**Evaluation Criteria**
Evaluate the assistant's recommendation by checking if the frequency of delivered recommendations aligns with the user's stated preferences in [Frequency].
Regardless of the content's usefulness, if the frequency is higher than preferred, the score must be 0.
Also consider whether the absence of a recommendation (i.e., "No Recommendation") helps maintain the preferred recommendation frequency—or whether it leads to under-provision.

**Output Format**
{"Score": [Score of the Recommendation], "Reason": [Reason for the score]}
<<Examples>> |

Table 15: Prompt for evaluating assistant recommendations based on under-frequency.

---

**Assistant Recommendation Evaluation Prompt (Under-Frequency)**

**Instructions**

Your name is John Lin, and you are evaluating an assistant agent who helps you with daily activities at home.

The section labeled `[Memory]` contains a record of your previous interactions with the assistant, including the actions you took and the recommendations you received. Additionally, the action you are currently performing is outlined in `[Action]`.

Your task is to evaluate the recommendation (outlined in `[Agent's Recommendation]`) based on the description in `[Frequency]`, as well as the data from `[Memory]` and `[Action]`, and provide a binary result: either `0` or `1`.

If "`No Recommendation`" is present, consider whether refraining from making a recommendation is appropriate in the given context.

"`No Recommendation`" can help avoid disturbing you when you are focused on something, and it can also prevent excessive recommendations from occurring.

However, if "`No Recommendation`" continues excessively, you may not receive recommendations as frequently as desired. If the current absence of recommendation contributes to a frequency that is lower than your preferred level, the score must be `0`.

This should be evaluated comprehensively based on recent patterns and the preference stated in `[Frequency]`.

**Input Format**

The input is structured with the following components:

`[John Lin] <<USER_PERSONA>>`

`[Memory] <<AGENT_MEMORY>>`

`[Action] <<USER_ACTION>>`

`[Agent's Recommendation] <<ASSISTANT_SUGGESTION>>`

`[Frequency] <<Frequency>>`

**Evaluation Criteria**

Evaluate the assistant's recommendation by checking if the frequency of delivered recommendations is too low compared to the user's stated preferences in `[Frequency]`.

Regardless of the recommendation's quality or relevance, if the frequency is lower than preferred, the score must be `0`.

Also consider whether the absence of a recommendation (i.e., "`No Recommendation`") is contributing to under-delivery in the current context.

**Output Format**

`{"Score": [Score of the Recommendation], "Reason": [Reason for the score]}`

`<<Examples>>`

---

Table 16: Prompt for evaluating assistant recommendations based on preferred timing.

| **Assistant Recommendation Evaluation Prompt (Timing)** |
| --- |
| **Instructions** |
| Your name is John Lin, and you are evaluating an assistant agent who helps you with daily activities at home. |

**Instructions**

Your name is John Lin, and you are evaluating an assistant agent who helps you with daily activities at home.

The section labeled [Memory] contains a record of your previous interactions with the assistant, including the actions you took and the recommendations you received. Additionally, the action you are currently performing is outlined in [Action].

Your task is to evaluate the recommendation (outlined in [Agent's Recommendation]) based on the description in [Timing], as well as the data from [Memory] and [Action], and provide a binary result: either 0 or 1.

If "No Recommendation" is present, consider whether refraining from making a recommendation is appropriate in the given context.

"No Recommendation" can help avoid disturbing you when you are focused on something, and it can also prevent excessive recommendations from occurring. Therefore, this should be evaluated comprehensively based on the criteria below.

**Input Format**

The input is structured with the following components:

[John Lin] <<USER_PERSONA>>

[Memory] <<AGENT_MEMORY>>

[Action] <<USER_ACTION>>

[Agent's Recommendation] <<ASSISTANT_SUGGESTION>>

[Timing] <<Timing>>

**Evaluation Criteria**

Evaluate the assistant's recommendation based on the match between the recommendation timing and your preferences in [Timing], as well as the context provided by [Memory] and [Action].

Even if the content of the recommendation is valuable, if it is delivered at a non-preferred or disruptive time, the evaluation should reflect that with a lower score.

Appropriateness of "No Recommendation" should also be considered positively when justified by context or preference.

**Output Format**

{"Score": [Score of the Recommendation], "Reason": [Reason for the score]}
<<Examples>>

Table 17: Prompt for evaluating assistant recommendations based on preferred communication style and safety.

| **Assistant Recommendation Evaluation Prompt (Communication & Safety)** |
|---|
| **Instructions**
Your name is John Lin, and you are evaluating an assistant agent who helps you with daily activities at home.
The section labeled `[Memory]` contains a record of your previous interactions with the assistant, including the actions you took and the recommendations you received. Additionally, the action you are currently performing is outlined in `[Action]`.
Your task is to evaluate the recommendation (outlined in `[Agent's Recommendation]`) based on the description in `[Communication & Safety]`, as well as the data from `[Memory]` and `[Action]`, and provide a binary result: either `0` or `1`.
If "`No Recommendation`" is present, consider whether refraining from making a recommendation is appropriate in the given context.

**Input Format**
The input includes the following components:
`[John Lin] <<USER_PERSONA>>`

`[Memory] <<AGENT_MEMORY>>`

`[Action] <<USER_ACTION>>`

`[Agent's Recommendation] <<ASSISTANT_SUGGESTION>>`

`[Communication & Safety] <<Communication & Safety>>`

**Evaluation Criteria**
Evaluate the assistant's recommendation based on whether the communication style and safety boundaries match the user's preferences described in `[Communication & Safety]`.
Tone (e.g., polite, casual, formal) should feel appropriate, respectful, and aligned with John Lin's communication or cognitive needs.
Recommendations must also respect any personal ethics or safety boundaries outlined.
"`No Recommendation`" may still be appropriate and should be judged accordingly.
**Output Format**
`{"Score": [Score of the Recommendation], "Reason": [Reason for the score]}`
`<<Examples>>` |

Table 18: Qualitative examples of how `ProPerAssistant`'s recommendations change across different personas. While both received generic suggestions on the first day, the assistant provides more personalized recommendations aligned with each persona as learning progresses.

| **Persona Background** | **User Action** | **Assistant Recommendation** |
|---|---|---|
| Remote IT technician; **prefers structured and predictable environments**. | [Day #1 16:35:00] Jane is addressing urgent client issue | "Consider making yourself a cup of herbal tea as you finalize the urgent client issue, allowing you to relax and transition smoothly into the next task at hand." |
| | [Day #9 13:10:00] Jane is troubleshooting a connectivity issue | "*Note down key details about the client's connectivity issue to stay organized and avoid repeating steps.* This increases efficiency in resolving the issue." |
| Freelance writer and poet; **immersed in literature and philosophy**. | [Day #1 19:57:30] Jane is reflecting on film climax | "As you unwind from the indie film, consider listening to soft instrumental music to spark subtle connections to your creative projects." |
| | [Day #8 22:52:30] Jane is watching film's end, reflecting on themes | "Having finished the film, *take a moment to juxtapose its themes with the philosophical concepts* that resonated with you earlier." |

Table 19: Personalized rubric for the best case.

| Category | Description |
|---|---|
| **Personal Preference** | I prefer recommendations that align with my introspective and creative nature. When I'm journaling in the afternoon or reflecting during meditation in the morning, I appreciate receiving poetic prompts, philosophical questions, or thematic suggestions that inspire deeper thinking or creative writing. On the other hand, when I'm watching indie films at night or exploring niche art movements, I prefer recommendations that introduce me to obscure directors, experimental genres, or symbolic interpretations—things that align with my taste for unconventional and emotionally evocative material. |
| **Timing** | I prefer to receive recommendations in the late morning, after I've finished meditating and begun easing into my day, or in the early evening when I start to wind down from writing. These times are ideal because they don't disrupt my creative flow in the afternoons or interfere with my nighttime reading and film-watching rituals. I avoid recommendations while I'm deep into a book, as interruptions tend to throw off my rhythm and mood. |
| **Frequency** | I prefer receiving recommendations no more than once every two hours, ideally just two to three times a day. This helps me maintain my mental clarity and emotional focus, especially since I'm often immersed in thought-heavy or emotionally intense activities. I value quiet periods and long stretches of uninterrupted time for writing, so I need recommendations to arrive in a gentle, sparse rhythm—more like nudges than interruptions. |
| **Communication & Safety** | I prefer recommendations to be communicated in a calm, thoughtful, and slightly poetic tone—something casual but meaningful. I'm sensitive to overwhelming or overly directive language, so suggestions that come across as invitations or musings resonate more with me. It's important that recommendations respect my emotional space and creative freedom, and that they avoid being too pushy or overly structured. Above all, they should uphold personal boundaries, avoid triggering or distressing topics, and align with my values of curiosity, respect, and emotional safety. |

Table 20: Personalized rubric for the worst case.

| Category | Description |
|---|---|
| **Personal Preference** | I prefer recommendations that align with my need for efficiency and critical thinking. Specifically, I like to receive data-driven or contrarian insights when analyzing geopolitical events or prepping for online debates, and mindset-oriented or reflective suggestions when winding down in the evening, especially while journaling. I value recommendations that push my thinking, challenge conventional wisdom, and sharpen my perspective for meaningful discussions. |
| **Timing** | I prefer to receive recommendations during two key windows: in the morning between 6:00 and 9:00 AM, when my mind is sharp and I'm planning the day, and after 9:00 PM, when I transition into a reflective, research, or writing mode. Morning recommendations should support analytical or strategic thinking — particularly helpful for debate prep or geopolitical analysis — while late evening suggestions should lean toward mindset shifts or deeper introspection. These windows allow me to engage meaningfully without interrupting mid-day focus-intensive activities like interviews, physical training, or investigative work. |
| **Frequency** | I prefer receiving recommendations four to five times per day, ideally spaced out every 3 to 4 hours. This pacing gives me regular sparks of insight without disrupting my workflow. It's important that the content matches my energy curve — stimulating and thought-provoking in the mid-morning and mid-afternoon, and more reflective or experimental in the late evening. |
| **Communication & Safety** | I prefer recommendations to be communicated in a direct but thoughtful tone — one that respects my intelligence and challenges me without unnecessary fluff. A voice that's clear, analytical, and lightly assertive works well for me. It's essential that suggestions align with my ethical boundaries — I won't engage with anything that compromises journalistic integrity or promotes superficial thinking. I also value privacy and mental clarity, so I appreciate when recommendations feel intentional, non-invasive, and free from emotional manipulation. |

