# OpenReview forum: "ProPerSim: Developing Proactive and Personalized AI Assistants through User-Assistant Simulation"
_ICLR.cc/2026/Conference — ICLR 2026 Poster_

### Official Review · Reviewer_b3zB · 2025-10-22

**Soundness:** 2
**Presentation:** 3
**Contribution:** 3
**Rating:** 6
**Confidence:** 3

**Summary:**

This paper introduces ProPerSim, a simulation framework for developing AI assistants that integrate proactivity and personalization in home environments. It also presents ProPerAssistant, a retrieval-augmented AI assistant trained using ProPerSim to adapt its strategies based on user feedback. ProPerAssistant demonstrates improved user satisfaction across 32 diverse personas, highlighting its ability to adapt to individual preferences and contexts. This work bridges the gap between personalized and proactive AI systems, advancing the development of context-aware, user-centric recommendations.

**Strengths:**

1. The paper addresses an important gap by combining personalization and proactivity, offering a new perspective on AI assistant design.

2. The simulation framework is well-structured and supports realistic evaluation through diverse user personas and rich feedback loops.

3. ProPerAssistant demonstrates measurable improvements in user satisfaction, showcasing its practical applicability and potential for real-world integration.

**Weaknesses:**

1. The dataset mentioned in the paper is neither provided in the supplementary materials nor shared with an anonymous link. This raises concerns about whether the dataset will be made publicly available in the future.

2. While the paper claims to integrate proactivity and personalization as its motivation, the focus of both the dataset and the proposed method appears to be predominantly on personalization. The treatment of proactivity seems rigid, as it only involves providing suggestions at fixed time intervals rather than adapting to users' behaviors. Such an approach lacks flexibility and does not align well with real-world user scenarios.

3. The paper includes comparisons with only a limited set of baselines, specifically a few variations of ProPerAssistant. However, it does not compare the proposed method against other existing personalization baselines. This omission raises concerns about the effectiveness of the proposed approach.

4. The authors report performance results based on only one type of large language model (LLM) and do not conduct ablation studies on other LLMs. This raises doubts about the generalizability of their method and dataset across different models.

**Questions:**

See weakness.

---

> ### Author Response · Authors · 2025-11-21
> **Response to Reviewer b3zB (1/2)**
>
> ```W1. The dataset mentioned in the paper is neither provided in the supplementary materials nor shared with an anonymous link. This raises concerns about whether the dataset will be made publicly available in the future.```
>
> Thank you for your valuable feedback. We apologize for the confusion regarding data availability. The anonymized repository can be accessed at: https://anonymous.4open.science/r/ProPerSim
>
> ```2. While the paper claims to integrate proactivity and personalization as its motivation, the focus of both the dataset and the proposed method appears to be predominantly on personalization. The treatment of proactivity seems rigid, as it only involves providing suggestions at fixed time intervals rather than adapting to users' behaviors. Such an approach lacks flexibility and does not align well with real-world user scenarios.```
>
> We appreciate the reviewer’s thoughtful comment regarding the balance between personalization and proactivity in ProPerSim. We would like to respectfully clarify, however, that our approach is not intended to prioritize personalization at the expense of proactivity. Our time-driven formulation is a deliberate design choice aimed at generalizing proactivity beyond the event-triggered paradigm that is common in prior work. At each timestep, the assistant explicitly decides whether to intervene or not, so proactivity is modeled as an adaptive policy decision rather than a fixed schedule. Importantly, the assistant is not required to issue recommendations at every interval; instead, through preference learning, it autonomously adjusts the timing and frequency of interventions based on user feedback.
>
> In addition, proactivity is directly reflected in our evaluation. Two of our four criteria—Timing and Frequency—are specifically designed to operationalize proactive behavior. Moreover, the user’s evaluative state encodes temporal context, recent activities, and interaction history, enabling the assistant to make dynamic and context-sensitive decisions about whether and when to act. Taken together, these components support flexible proactivity that better aligns with real-world user scenarios.

---

> ### Author Response · Authors · 2025-11-21
> **Response to Reviewer b3zB (2/2)**
>
> ```W3. The paper includes comparisons with only a limited set of baselines, specifically a few variations of ProPerAssistant. However, it does not compare the proposed method against other existing personalization baselines. This omission raises concerns about the effectiveness of the proposed approach.```
>
> ```W4. The authors report performance results based on only one type of large language model (LLM) and do not conduct ablation studies on other LLMs. This raises doubts about the generalizability of their method and dataset across different models.```
>
> Thank you for these thoughtful comments. Regarding the limited set of baselines, most representative personalization methods in the literature achieve personalization by retrieving user-specific information, such as profiles, interaction histories, or search logs, and then conditioning an LLM on that retrieved context [1,2,3,4]. In line with this common paradigm, our baselines were designed as RAG-based personalization methods that leverage users’ actions and received recommendations. The ProPerAssistant variants included in our experiments therefore correspond to retrieval-augmented personalization baselines that are closely aligned with prior approaches, but implemented within the same experimental framework for fair comparison.
>
> To further address concerns about baseline diversity and model generalizability, we additionally implemented a baseline using a recent state-of-the-art reasoning model. Specifically, under the ARS memory configuration, we replaced the default LLM with a more advanced reasoning model (GPT-5). Due to budget constraints, we conducted this experiment on 10 personas. The results are summarized in the table below and indicate that our method remains effective under a stronger backbone model. We will incorporate these results and provide a detailed discussion in the revised manuscript.
>
> In addition, following another reviewer’s suggestion, we are currently running experiments on a ProPerAssistant variant trained with KTO instead of DPO. We will include the outcomes of this study, along with an expanded discussion, in the final version of the paper.
>
> Table. Comparison of GPT-5 and Llama 3.3 70B in the ARS Memory Setting
>
> | Persona | gpt-5 Day 1 | gpt-5 Day 14 | Llama 3.3 70b Day 1 | Llama 3.3 70b Day 14 |
> |---------|-------------|--------------|----------------------|-----------------------|
> | 1       | 2.81        | 2.98         | 2.61                 | 2.64                  |
> | 2       | 3.41        | 3.28         | 2.61                 | 2.53                  |
> | 3       | 2.58        | 2.77         | 2.23                 | 2.37                  |
> | 4       | 2.78        | 2.83         | 2.51                 | 2.44                  |
> | 5       | 2.84        | 2.78         | 2.41                 | 2.45                  |
> | 6       | 2.68        | 3.02         | 2.38                 | 2.51                  |
> | 7       | 2.83        | 3.09         | 2.91                 | 2.59                  |
> | 8       | 2.86        | 2.93         | 2.67                 | 2.41                  |
> | 9       | 2.99        | 3.29         | 2.28                 | 2.47                  |
> | 10      | 2.68        | 2.70         | 2.69                 | 2.58                  |
>
>
>
> [1] Salemi, Alireza, et al. "Lamp: When large language models meet personalization." Proceedings of the 62nd Annual Meeting of the Association for Computational Linguistics (Volume 1: Long Papers). 2024.
>
> [2] Salemi, Alireza, Surya Kallumadi, and Hamed Zamani. "Optimization methods for personalizing large language models through retrieval augmentation." Proceedings of the 47th International ACM SIGIR Conference on Research and Development in Information Retrieval. 2024.
>
> [3] Shi, Teng, et al. "Retrieval augmented generation with collaborative filtering for personalized text generation." Proceedings of the 48th International ACM SIGIR Conference on Research and Development in Information Retrieval. 2025.
>
> [4] Li, Xiaopeng, et al. "A survey of personalization: From rag to agent." arXiv preprint arXiv:2504.10147 (2025).

---

> > ### Author Response · Authors · 2025-11-29
> > **KTO Experimental Results**
> >
> > For ProPerAssistant using KTO, we ran experiments on seven personas, which was the maximum we could complete within the constrained rebuttal timeline. On average, the performance showed no significant difference compared to DPO, and we will include these results in the revised manuscript.
> >
> > Table 1. Comparison between training with KTO and training with DPO in ProPerAssistant
> >
> > | Persona | KTO Day 1 | KTO Day 10 | DPO Day 1 | DPO Day 10 |
> > |---------|-------------|--------------|----------------------|-----------------------|
> > | 1       | 2.48        | 2.93         | 2.49                 | 3.39                  |
> > | 2       | 2.55        | 3.44         | 2.59                 | 3.55                  |
> > | 3       | 2.08        | 2.59         | 1.93                 | 2.30                  |
> > | 4       | 2.24        | 2.84         | 2.37                 | 2.96                  |
> > | 5       | 2.10        | 2.94         | 2.13                 | 2.55                  |
> > | 6       | 2.22        | 3.24         | 2.36                 | 3.15                  |
> > | 7       | 2.50        | 3.56         | 2.33                 | 3.84                  |

---

### Official Review · Reviewer_PbEY · 2025-10-27

**Soundness:** 4
**Presentation:** 3
**Contribution:** 3
**Rating:** 6
**Confidence:** 3

**Summary:**

This paper investigates the proactive personalization problem in the conversational ai scenario. Due to the rare study in this track, this work proposes ProPerSim to simulate this scenario and further propose ProPerAssistant which coupled with a memory system and a preference-aligned training strategy to proactively generate personalized interventions for user.

**Strengths:**

* The research scenario is very interesting and to the community’s focus.

* The proposed ProPerSim simulation is concrete and consider personality, environment, user’ memory.

* The proposed ProPerAssistant learns from user feedback via intermediate state and user preference is novel.

**Weaknesses:**

Some minors from practical:

* The baseline selections are not convincing, the ProPerSim would of course outperforms than no memory, AR memory and ARS memory since ProPerSim are trained based on these configurations and contain them all. The comparisons here are more like ablation. Authors may consider different preference alignment comparisons etc.

* The experiments lack the evaluation of intervention. Since it’s proactive personalization, it would be straightforward to see what the rate of successfully proper interventions is.

* Latency/efficiency analysis on the daily data generation should be included.

**Questions:**

* How does adopting a purely time-based framing of proactivity potentially limit the assistant's ability to respond to specific user-triggered events (which prior work used) that might require instantaneous intervention, rather than waiting for the next predetermined timestep?

* Is the challenge of improving performance for these complex personas fundamentally attributable to the memory system's inability to retain the necessary temporal and content granularity required to meet these highly specific demands, given that the memory relies heavily on time block summarization?

---

> ### Author Response · Authors · 2025-11-21
> **Response to Reviewer PbEY (1/4)**
>
> ```W1. The baseline selections are not convincing, the ProPerSim would of course outperforms than no memory, AR memory and ARS memory since ProPerSim are trained based on these configurations and contain them all. The comparisons here are more like ablation. Authors may consider different preference alignment comparisons etc.```
>
> Thank you for this helpful feedback. We agree that the “no memory” and “AR memory” settings can be viewed as ablation-style baselines. However, we would like to clarify that ARS memory is not a weaker setting in terms of information available at inference time. Unlike ProPerAssistant, which relies on preferences learned through preference modeling during training, ARS memory provides the explicit numerical scores for each action–recommendation pair directly in-context at inference. This comparison therefore allows us to examine whether providing explicit score feedback in-context during inference can be more effective than learning preferences through preference modeling. As shown in Figure 4, ARS memory achieves higher scores on Day 1; however, over subsequent days, ProPerAssistant consistently outperforms ARS memory.
>
> Following your suggestion, we also added two additional baselines.
>
>  1. ARS memory with a stronger reasoning model. To verify whether ARS memory can better exploit the in-context preference information when paired with a more advanced model, we replaced the default LLM with a stronger reasoning model (GPT-5) under the ARS memory configuration. The results are summarized in the table below and indicate that our method remains effective under a stronger backbone model. Due to budget constraints, we ran this experiment on 10 personas, and we summarize the results in the table below.
>
>  2. ProPerAssistant trained with KTO instead of DPO. We are currently running this experiment and will report the results as soon as they are available.
>
> Table 1. Comparison of GPT-5 and Llama 3.3 70B in the ARS Memory Setting
>
> | Persona | GPT-5 Day 1 | GPT-5 Day 14 | Llama 3.3 70b Day 1 | Llama 3.3 70b Day 14 |
> |---------|-------------|--------------|----------------------|-----------------------|
> | 1       | 2.81        | 2.98         | 2.61                 | 2.64                  |
> | 2       | 3.41        | 3.28         | 2.61                 | 2.53                  |
> | 3       | 2.58        | 2.77         | 2.23                 | 2.37                  |
> | 4       | 2.78        | 2.83         | 2.51                 | 2.44                  |
> | 5       | 2.84        | 2.78         | 2.41                 | 2.45                  |
> | 6       | 2.68        | 3.02         | 2.38                 | 2.51                  |
> | 7       | 2.83        | 3.09         | 2.91                 | 2.59                  |
> | 8       | 2.86        | 2.93         | 2.67                 | 2.41                  |
> | 9       | 2.99        | 3.29         | 2.28                 | 2.47                  |
> | 10      | 2.68        | 2.70         | 2.69                 | 2.58                  |

---

> > ### Author Response · Authors · 2025-11-29
> > **KTO Experimental Results**
> >
> > For ProPerAssistant using KTO, we ran experiments on seven personas, which was the maximum we could complete within the constrained rebuttal timeline. On average, the performance showed no significant difference compared to DPO, and we will include these results in the revised manuscript.
> >
> > Table 1. Comparison between training with KTO and training with DPO in ProPerAssistant
> >
> > | Persona | KTO Day 1 | KTO Day 10 | DPO Day 1 | DPO Day 10 |
> > |---------|-------------|--------------|----------------------|-----------------------|
> > | 1       | 2.48        | 2.93         | 2.49                 | 3.39                  |
> > | 2       | 2.55        | 3.44         | 2.59                 | 3.55                  |
> > | 3       | 2.08        | 2.59         | 1.93                 | 2.30                  |
> > | 4       | 2.24        | 2.84         | 2.37                 | 2.96                  |
> > | 5       | 2.10        | 2.94         | 2.13                 | 2.55                  |
> > | 6       | 2.22        | 3.24         | 2.36                 | 3.15                  |
> > | 7       | 2.50        | 3.56         | 2.33                 | 3.84                  |

---

> ### Author Response · Authors · 2025-11-21
> **Response to Reviewer PbEY (2/4)**
>
> ```W2. The experiments lack the evaluation of intervention. Since it’s proactive personalization, it would be straightforward to see what the rate of successfully proper interventions is.```
>
> Thank you for the insightful comment. In response, we calculated the proportion of all interventions that received a rating of 3 or higher on a 4-point scale for each day from Day 1 to Day 14. The proportion increased from 51.06% on Day 1 to 71.51% on Day 14, indicating an improvement of about 20 percentage points in the rate of successful interventions. We will add a discussion of this analysis to the revised manuscript.
>
> Table 2. Proportion of Successful Interventions (Rating ≥ 3) from Day 1 to Day 14 (%)
>
> | 1     | 2     | 3     | 4     | 5     | 6     | 7     | 8     | 9     | 10    | 11    | 12    | 13    | 14    |
> |-------|-------|-------|-------|-------|-------|-------|-------|-------|-------|-------|-------|-------|-------|
> | 51.06 | 53.99 | 59.01 | 63.91 | 64.87 | 65.37 | 65.50 | 69.51 | 68.51 | 66.97 | 65.57 | 70.55 | 68.88 | 71.51 |
>
> ```W3. Latency/efficiency analysis on the daily data generation should be included.```
>
> Thank you for the insightful comment. In our experiments, the overall runtime varies depending on the chosen time-step configuration and the underlying hardware. For example, if efficient training is the priority, setting the time step to 20 minutes allows a 14-day simulation to be completed in roughly one day on a single A100 GPU. Using an H100 GPU can further reduce this runtime. In contrast, for our main experiments, we intentionally set a smaller time step of 2.5 minutes to enable the assistant to interact in near real time. Under this setting, a single 14-day experiment required approximately 10 days of computation.
>
> We will incorporate a detailed discussion of this latency and efficiency analysis into the revised manuscript.

---

> ### Author Response · Authors · 2025-11-21
> **Response to Reviewer PbEY (3/4)**
>
> ```Q1. How does adopting a purely time-based framing of proactivity potentially limit the assistant's ability to respond to specific user-triggered events (which prior work used) that might require instantaneous intervention, rather than waiting for the next predetermined timestep?```
>
> Thank you for this valuable question. In our simulation, the user agent and the assistant operate on the same discrete timestep. Therefore, whenever the user’s state or behavior changes at a given timestep, the assistant can issue a recommendation within that same timestep. In other words, the environment does not allow a user-triggered change to occur “between” predetermined steps without an opportunity for the assistant to react.
>
> We agree that the distinction between a purely time-based framing and an event-triggered framing is conceptually important. Our approach uses time steps primarily as a modeling convenience. If the timestep is chosen to be sufficiently small (e.g., 1ms), the time-based formulation can approximate instantaneous, event-driven intervention: any user-triggered event will be captured at the next (very near) step, enabling the assistant to respond effectively in near real time. Thus, while our framework is time-indexed, it does not inherently prevent timely responses to abrupt user-triggered events; rather, responsiveness is governed by the timestep granularity.

---

> ### Author Response · Authors · 2025-11-21
> **Response to Reviewer PbEY (4/4)**
>
> ```Q2. Is the challenge of improving performance for these complex personas fundamentally attributable to the memory system's inability to retain the necessary temporal and content granularity required to meet these highly specific demands, given that the memory relies heavily on time block summarization?```
>
> Thank you for the thoughtful question. We don’t believe the difficulty in improving performance for complex personas comes primarily from a lack of memory capacity or from time-block summarization. The amount of information in the user’s evaluative state is comparable to assistant’s internal state, so the memory system itself is not the main bottleneck.
>
> A more important issue is the limited form of the training signal. The user evaluates recommendations using four criteria, but this multi-dimensional judgment is compressed into a single integer score (or a preference pair). Crucially, the user does not tell the assistant which rubric(s) drove a high or low score. As a result, the assistant must infer the underlying criteria solely from scalar scores or pairwise preferences. This compression makes learning harder in general, and the problem becomes especially pronounced for complex personas where the evaluation criteria interact in more nuanced ways.
>
> In principle, this could be alleviated by providing richer feedback from the user agent—for example, alongside scores, giving short explanations of why each recommendation was rated as it was, and using those explanations as additional supervision. However, in realistic human–assistant settings, we cannot rely on users to consistently provide such detailed rationales. Understanding and closing this gap between idealized training signals and practical user behavior remains an important direction for future work.

---

### Official Review · Reviewer_6KAu · 2025-11-06

**Soundness:** 3
**Presentation:** 3
**Contribution:** 3
**Rating:** 8
**Confidence:** 3

**Summary:**

In this work, the authors introduce a simulation framework and benchmark for developing AI assistants that are both proactive and personalized. The framework models a simulated home environment where a user agent performs daily routines, with multiple distinct user personas and realistic activity patterns generated using GPT-4o. The assistant’s objective is to determine when and what actions to recommend to the user. To address this, the authors propose a retrieval-augmented, preference-learning assistant that maintains a memory of past interactions, generates multiple recommendation candidates, and learns from user feedback through preference optimization. Extensive experiments demonstrate that this approach substantially improves user satisfaction across a wide range of simulated personas.

**Strengths:**

The main strength of this paper lies in its timely motivation to create assistants that are both proactive and personalized. The authors create extremely controlled agent setup to study the design of such systems. Under this setup, they consider multiple constraints that the assistant need to adhere to, creating an extremely rich setting to ablate different scenarios. The authors also validate their proposed setup with a human evaluation study, which increases validity of their claims about naturalness and persona alignment.

In addition, the authors propose intuitive system that builds on 2 key components: (a) structured memory storing past experiences, and (b) DPO training based on user feedback. Through extensive evaluation, their proposed system shows consistent improvement across all their persons. Overall, this is an extremely strong paper that will be of great interest to the scientific community.

**Weaknesses:**

I don't see many weaknesses with the paper. In fact, the authors have gone to extreme details in designing their system.

I have a few questions with the experimental setup.

(a) The definition of proactivity is narrow. Under the current definition, proactivity is reduced to timing control, not anticipation of latent goals or needs of the user. This is because the authors define it as whether an action should be taken at a particular time. However, by proactivity, I would also define it in terms whether an action can return a reward faster or change the trajectory of decisions on the user. How can that be modeled under the current framework?

(b) Noisy or delayed feedback: One key missing component in the current setup is that it assumes almost perfect feedback from the user always. User Preferences can be noisy or even delayed. How would such noise be incorporated into such a setup? And how would the proposed system change under presence of such noise?

(c) LLM-as-a-judge is performed with Gemini 2.0 Flash. The rubric used in table 6 is extremely broad. How would the results change when the evaluation is performed with a different LLM?

**Questions:**

Please check my questions above.

---

> ### Author Response · Authors · 2025-11-21
> **Response to Reviewer 6KAu (1/3)**
>
> ```W(a) The definition of proactivity is narrow. Under the current definition, proactivity is reduced to timing control, not anticipation of latent goals or needs of the user. This is because the authors define it as whether an action should be taken at a particular time. However, by proactivity, I would also define it in terms whether an action can return a reward faster or change the trajectory of decisions on the user. How can that be modeled under the current framework?```
>
> Thank you for the insightful comment. We agree that proactivity can be defined more broadly—not only as choosing the right timing, but also as anticipating a user’s latent goals or needs and helping shape their longer-term decision trajectory. If we adopt this richer notion of proactivity, our evaluation framework can be extended by incorporating delayed rewards.
>
> Concretely, we can (1) allow latent goals to be specified or updated by the user at certain points, and (2) assign delayed rewards based on the user’s eventual satisfaction with respect to those goals when the assistant acts proactively. In other words, while we continue to provide immediate rewards at each time step, we would also add a delayed reward signal tied to long-term outcomes.
>
> For example, alongside our current rubric (Personal Preference, Timing, Frequency, Communication & Safety), we could introduce a new dimension such as Long-term Satisfaction. If a user’s latent long-term goal is financial stability or expense management, then a proactive recommendation like “Some of the ingredients you often buy are on sale this week” during recipe search might not be strongly relevant to the immediate context, yet it could meaningfully support the user’s long-term objective. The user could then evaluate such suggestions after some time, and we could assign delayed rewards accordingly.
>
> We will add a discussion of this possible extension—particularly the role of delayed rewards for modeling latent goals—into the revised manuscript.

---

> ### Author Response · Authors · 2025-11-21
> **Response to Reviewer 6KAu (2/3)**
>
> ```W(b) Noisy or delayed feedback: One key missing component in the current setup is that it assumes almost perfect feedback from the user always. User Preferences can be noisy or even delayed. How would such noise be incorporated into such a setup? And how would the proposed system change under presence of such noise?```
>
> Thank you for the insightful comment. Regarding delayed feedback, we address this in our response to the previous question.
> As you point out, real users are unlikely to provide perfectly accurate or immediate feedback. Our setup can be naturally extended to model such imperfections in several ways. For instance, we could inject noise by randomly perturbing the scores for a subset of feedback signals before passing them to the assistant. When the noise level is modest, we expect that running the simulation over a sufficiently long horizon would still allow the system to recover the user’s underlying preferences. However, under high noise, the system would likely require substantially more data and longer training to achieve convergence.
>
> We will add a dedicated discussion of noisy to the main text, and we plan to extend our codebase to explicitly support this alternative setting.

---

> ### Author Response · Authors · 2025-11-21
> **Response to Reviewer 6KAu (3/3)**
>
> ```W(c) LLM-as-a-judge is performed with Gemini 2.0 Flash. The rubric used in table 6 is extremely broad. How would the results change when the evaluation is performed with a different LLM?```
>
> We apologize for any confusion caused by this part. The rubric candidates shown in Table 6 are indeed extremely broad because they were compiled before filtering via the survey. The actual rubrics we used for evaluation are based on the four criteria illustrated in Tables 8–11.
>
> To select an appropriate LLM evaluator, we experimented with a variety of models. We found that smaller models such as GPT-4o-mini and GPT-4.1-mini were not able to provide sufficiently consistent evaluations. In contrast, GPT-4o, Gemini 2.0 Flash, Gemini 2.5 Flash, and GPT-o4-mini all produced stable and consistent judgments. Among these, we chose Gemini 2.0 Flash primarily because it was the most cost-effective option.
>
> We also considered local reasoning models as potential evaluators, but due to latency issues, we ultimately restricted our candidates to API-based LLMs.

---

### Official Review · Reviewer_z9Wn · 2025-11-07

**Soundness:** 2
**Presentation:** 3
**Contribution:** 2
**Rating:** 4
**Confidence:** 3

**Summary:**

This paper created a benchmark for simultaneously evaluating the proactivity and personalization of an agent. The benchmark is a simulated environment where a user which is powered by an advanced LLM conducts daily activities, simulating human behaviors. At the same time, an agent observes the activities of the user and decides whether or not to interact with the user by providing recommendations. The agent will receive a score of its recommendation based on the user's personalized rubric evaluated by an LLM. The users are created to cover a broad set of persona. Besides the benchmark, the authors also provide an agent that performs well under this simulated environment by utilizing preference alignment and retrieval-augmented generation.

**Strengths:**

1. This paper considers the problem of benchmarking both proactivity and personalization of an agent. It can augment the existing evaluations which mainly focus on either proactivity or personalization.

2. The presentation of this paper is clear and easy to follow.

**Weaknesses:**

1. The definition of proactivity is too narrow. In this proposed benchmark, proactivity is mainly described as the ability to initiate the interaction with the user by providing recommendations. On the contrary, previous works on benchmarking proactivity covers a broad range of tasks including coding, writing, and daily life scenarios. Also, the quantification/evaluation of proactivity seems to be mixed into personalization. For example, how can one infer from the evaluation results that the agent is proactive but bad at personalization?

2. The proposed ProPerAssistant can be regarded as a baseline of this benchmark instead of a contribution, since there is no new innovation in this agent which mainly based on memory retrieval and preference optimization. For example, the statement at line 406 "Notably, ProPerAssistant achieves this without relying on computationally expensive reinforcement
learning methods, instead leveraging an efficient DPO-based approach to learn user preferences". This is the contribution of DPO. And I think the statement is a little bit misleading since the agent is trained with online DPO, which necessitates the collection of online trajectories, it is still expensive.

3. The benchmark is not practical to use. the simulation and evaluation are both powered by advanced LLMs which induces lots of API costs. And since the evaluation is based on LLM-as-a-judge. There can be bias or hallucination. Although in section 4.3, the authors conducted manual checking of the actions and evaluations, it is a limited checking and I still concern the reliability of the benchmark.

4. As an agent capable of actively engaging the interaction and infer the personality of a user based on interaction history, it should be able to generalize to users with other persona without re-training. But there is limited discussion (not sure the first paragraph in section 6.4 is under this setting?) on the generalization ability of ProPerAssistant.

**Questions:**

I also raised these questions in the weaknesses part. Please refer to that part for more contents.

1. How can one infer the separate performance of proactivity or personalization from the evaluation results? It seems low proactivity or bad personalization will decrease the score. I know the goal of this benchmark is to evaluate the joint performance. But it is also crucial to know what the agent is bad at currently for future improvement.

2. What is the setting of Adaption Across Diverse Personas in section 6.4? What's the training set of users and what's the evaluation set?

---

> ### Author Response · Authors · 2025-11-21
> **Response to Reviewer z9Wn (1/5)**
>
> ```W1-1. The definition of proactivity is too narrow. In this proposed benchmark, proactivity is mainly described as the ability to initiate the interaction with the user by providing recommendations. On the contrary, previous works on benchmarking proactivity covers a broad range of tasks including coding, writing, and daily life scenarios. ```
>
> Thank you for the valuable comment. We would like to clarify that our definition of proactivity is not intended to be narrow; rather, our benchmark specifically targets the development of a personal assistant in a home environment. Just as prior work evaluates agents that proactively make coding-related suggestions during coding tasks or writing-related suggestions during writing tasks, our goal is to design an assistant that provides appropriate and helpful recommendations to users in their everyday home life.
>
> If our current writing or title gives the impression that we are making a broader claim than intended, we will adjust the tone accordingly. We will explicitly emphasize that our benchmark is aimed at developing personal assistants for home environments. For example, we can revise the title to something like: “ProPerSim: Developing Proactive and Personalized AI Assistants through User–Assistant Simulation in Home Environments.”

---

> ### Author Response · Authors · 2025-11-21
> **Response to Reviewer z9Wn (2/5)**
>
> ```W1-2. Also, the quantification/evaluation of proactivity seems to be mixed into personalization. For example, how can one infer from the evaluation results that the agent is proactive but bad at personalization?```
>
> ```Q1. How can one infer the separate performance of proactivity or personalization from the evaluation results? It seems low proactivity or bad personalization will decrease the score. I know the goal of this benchmark is to evaluate the joint performance. But it is also crucial to know what the agent is bad at currently for future improvement.```
>
> We thank the reviewer for raising this important question. In our rubric, proactivity and personalization are reflected in different criteria. If we interpret proactivity as when and how the assistant chooses to initiate an action, then the rubric items on Timing and Frequency directly capture this dimension. If we interpret personalization as how well the assistant tailors the content to the user’s preferences, then Personal Preference and Communication & Safety indicate content-level personalization. Using this breakdown, one can inspect which criteria contributed to low performance—for example, poor Timing/Frequency but strong content alignment would indicate proactive miscalibration rather than weak personalization. Figure 7 of the paper further illustrates how these aspects change before and after training.
>
> More fundamentally, our position is that in realistic human–assistant interaction, proactivity cannot be meaningfully evaluated in complete isolation from personalization. To make this point concrete, consider two users with contrasting preferences for proactive behavior:
> 1. Sarah prefers that the assistant intervene sparingly—no more than three times a day, and only when she is mentally available (e.g., during idle moments).
> 2. Paul, in contrast, expects proactive support: he wants hourly meaningful suggestions and becomes dissatisfied if the assistant remains silent for too long.
>
> In such cases, whether the assistant is “appropriately proactive” depends entirely on the user’s individual preference profile. A behavior that constitutes ideal proactivity for Paul (frequent, assertive suggestions) would be perceived as intrusive or even annoying for Sarah, even if the underlying “initiative-taking” mechanism were identical. Thus, attempting to score proactivity independently from personalization would systematically mischaracterize what the user perceives as helpful initiative.
>
> From this perspective, the notion of an assistant that is “good at proactivity but bad at personalization” becomes practically incoherent: initiative that ignores user preferences—e.g., recommending dinner ideas every hour to a user who dislikes frequent suggestions—is not successful proactivity in any user-centric sense. Personalization is therefore not an optional add-on but a prerequisite for proactivity to be evaluated as effective rather than intrusive.
>
> For these reasons, ProPerSim is intentionally designed to assess the joint capability of proactive and personalized behavior, rather than producing fully disentangled scores. At the same time, our rubric-level decomposition still provides actionable diagnostic signals. For instance, if the assistant exhibits good Preference alignment but repeatedly violates Timing or Frequency, one can conclude that it understands the user’s tastes but lacks calibrated proactive behavior. This allows researchers to infer which component requires improvement, even without treating proactivity and personalization as independent axes.

---

> ### Author Response · Authors · 2025-11-21
> **Response to Reviewer z9Wn (3/5)**
>
> ```W2. The proposed ProPerAssistant can be regarded as a baseline of this benchmark instead of a contribution, since there is no new innovation in this agent which mainly based on memory retrieval and preference optimization. For example, the statement at line 406 "Notably, ProPerAssistant achieves this without relying on computationally expensive reinforcement learning methods, instead leveraging an efficient DPO-based approach to learn user preferences". This is the contribution of DPO. And I think the statement is a little bit misleading since the agent is trained with online DPO, which necessitates the collection of online trajectories, it is still expensive.```
>
> Thank you for this helpful feedback. Our intention was not to present ProPerAssistant as a core methodological contribution, but rather as a concrete baseline instantiation within ProPerSim that illustrates how a practical assistant can be learned using the benchmark. To avoid any confusion, we will revise the paper to describe ProPerAssistant explicitly as a baseline implementation rather than a proposed novel method.
>
> We also appreciate the point that online DPO still requires collecting online trajectories and is therefore not “cheap” in an absolute sense. Our use of the term efficient was meant in a relative sense: compared to typical online RL approaches such as PPO, which requires training an additional value function [1], or GRPO-style methods that often use multiple rollouts per prompt (e.g., 8–16) [2,3,4,5], online DPO can be less computationally expensive. That said, we agree the current wording may overstate the claim. We will revise the sentence to state more precisely that online DPO is less expensive than other online RL methods, or remove the efficiency claim altogether if the reviewer still finds it overreaching.
>
> [1] Schulman, John, et al. "Proximal policy optimization algorithms." arXiv preprint arXiv:1707.06347 (2017).
>
> [2] Shao, Zhihong, et al. "Deepseekmath: Pushing the limits of mathematical reasoning in open language models." arXiv preprint arXiv:2402.03300 (2024).
>
> [3] Guo, D., Yang, D., Zhang, H. et al. DeepSeek-R1 incentivizes reasoning in LLMs through reinforcement learning. Nature 645, 633–638 (2025). https://doi.org/10.1038/s41586-025-09422-z
>
> [4] Mroueh, Youssef. "Reinforcement Learning with Verifiable Rewards: GRPO's Effective Loss, Dynamics, and Success Amplification." arXiv preprint arXiv:2503.06639 (2025).
>
> [5] Bamba, Udbhav, et al. "XRPO: Pushing the limits of GRPO with Targeted Exploration and Exploitation." arXiv preprint arXiv:2510.06672 (2025).

---

> ### Author Response · Authors · 2025-11-21
> **Response to Reviewer z9Wn (4/5)**
>
> ```W3. The benchmark is not practical to use. the simulation and evaluation are both powered by advanced LLMs which induces lots of API costs. And since the evaluation is based on LLM-as-a-judge. There can be bias or hallucination. Although in section 4.3, the authors conducted manual checking of the actions and evaluations, it is a limited checking and I still concern the reliability of the benchmark.```
>
> We appreciate the reviewer’s concern regarding the practicality and reliability of a benchmark that relies on API-based LLMs for both simulation and evaluation. We agree that such dependence can raise cost issues, and that LLM-as-a-judge evaluations may introduce bias or hallucination. In our own development process, we initially aimed to use open-source models for both components, but latency constraints led us to adopt a faster API model instead. We expect, however, that as hardware and open-source reasoning models continue to improve, our framework will be able to support strong open-source evaluators without compromising efficiency.
>
> To directly address the reliability concerns, we conducted human evaluations at two levels:
>
> First (reported in Section 4.3 of the main paper), to validate the naturalness of the simulated daily lives and their alignment with target personas, we simulated an entire day for each of the 32 personas and asked human annotators to rate naturalness and persona alignment. Ten annotators participated, and each persona’s day was assessed by three annotators. Using a 0–10 Likert scale, the average naturalness score was 8.25 and persona alignment 8.02, indicating consistently high quality. Only 5.11% of generated actions were judged as incorrect. Importantly, these results were obtained across all 32 diverse personas, suggesting that bias or hallucination did not materially affect the simulated behaviors.
>
> Second (extended during rebuttal), the main paper already included a human study on the recommendation evaluator for three personas. During the rebuttal period, we expanded this same protocol to 16 additional personas, bringing the total to 19 personas evaluated by humans. Three annotators observed simulations and assessed recommendations across three time windows (morning, afternoon, night). Across the 19 personas, the evaluator obtained an average human-judged correctness of 90.97% with a standard deviation of 3.09, indicating strong agreement with human judgments. We are continuing this expansion to cover all 32 personas and will report the full results in the final version.
>
> Given these human studies—both the results already included in the paper and the newly expanded rebuttal evaluation—we believe the benchmark’s reliability is supported beyond “limited checking.” If we have misunderstood any part of your concern, or if there are additional questions we should clarify, we would be grateful if you could let us know. We are happy to provide further details.

---

> > ### Author Response · Authors · 2025-12-03
> > **Further Evidence for the Reliability of Our Evaluator**
> >
> > During the final stage of the rebuttal, we further extended the human evaluation of the recommendation evaluator from 19 to 26 personas. The evaluator achieved an average human-judged correctness of 90.83% with a standard deviation of 3.38, which is consistent with and further strengthens the reliability trends reported earlier. We are on track to finish all 32 personas and will include the complete results in the final version.

---

> ### Author Response · Authors · 2025-11-21
> **Response to Reviewer z9Wn (5/5)**
>
> ```W4. As an agent capable of actively engaging the interaction and infer the personality of a user based on interaction history, it should be able to generalize to users with other persona without re-training. But there is limited discussion (not sure the first paragraph in section 6.4 is under this setting?) on the generalization ability of ProPerAssistant.```
>
> Thank you for this thoughtful comment. In ProPerSim, the assistant is never given a user’s persona explicitly; it only observes the interaction history and the user’s current action. As a result, if the assistant is trained solely on interactions generated from a single persona, it will naturally absorb that persona’s preferences into its parameters. Deploying such a persona-specialized assistant to a user with substantially different preferences—without any retraining or adaptation—would therefore be expected to yield suboptimal behavior, simply because its prior over user preferences is mismatched.
>
> Accordingly, our goal in this paper is not to claim zero-shot transfer across qualitatively different personas. Instead, we focus on showing that, within a given simulated environment, ProPerAssistant can infer and adapt to a user’s latent state and preferences from interaction data. We agree that strong generalization to entirely new personas without additional learning is an important direction, but we see it as orthogonal to the main scope of this work and leave it for future research.
>
> If this interpretation does not fully capture your concern about generalization without retraining, we would greatly appreciate further clarification so that we can better address your point.
>
> ```Q2. What is the setting of Adaption Across Diverse Personas in section 6.4? What's the training set of users and what's the evaluation set?```
>
> We apologize for any confusion caused by this part. In the main text, “Adaptation Across Diverse Personas” refers to training an assistant for each persona and then evaluating it on the same persona it was trained on. To make this clearer, we will revise the subsection title to “Training on Diverse Personas”.

---

### Meta-Review · Area_Chair_Wh4v · 2026-01-08

**Summary:**

Reviewers agree the topic is important. Moreover, the simulated benchmark for proactive and personalized AI assistant is promising.
The main concerns driving a cautious accept are:
- proactivity definition is viewed as too time-indexed and not cleanly separable from personalization
- baseline breadth—comparisons look ablation-like and ProPerAssistant should be framed as a baseline rather than a key novelty
- practicality/reliability—heavy reliance on API LLMs and LLM-as-judge raises cost and bias concerns
- robustness/generalization across personas and LLM backbones is limited

I recommend Accept, but with the major caveat that many fixes are promised for the revision rather than reflected in the current manuscript.

**Reviewer Concerns:**

Rebuttal (partially) addressed:
- Proactivity definition is narrow / hard to disentangle from personalization (All reviewers): Narrows scope to home assistants; Argues rubric-level separation: Timing/Frequency vs Preference/Communication; Claims time-indexed steps approximate event-triggered behavior via step granularity.
- Baselines limited / comparisons look like ablations; ProPerAssistant novelty/positioning (PbEY, z9Wn, b3zB): Admits some settings are ablation-like; Commits to reframe ProPerAssistant explicitly as a baseline; Argues baselines are representative RAG-style personalization implemented within same framework.
- Practicality/cost + LLM-as-judge reliability (z9Wn, 6KAu): Expands human validation and reports high agreement; claims evaluator stability across several frontier judges; Gemini chosen for cost; Provides latency/efficiency discussion for simulation settings.
- Robustness/generalization across personas and LLM backbones (z9Wn, b3zB): Adds a stronger-backbone check (e.g., GPT-5 under ARS) and reports KTO vs DPO shows no significant change.

Still outstanding:
- Proactivity definition is narrow / hard to disentangle from personalization (z9Wn, b3zB): proactivity still feels timing-centric; disentanglement remains imperfect and partly philosophical; key improvements are mostly ``will revise''.
- Baselines limited (b3zB, PbEY): External/established personalization baselines are still not convincingly covered; some baseline additions are promised rather than in-paper.
- Robustness/generalization across personas and LLM backbones (z9Wn): Limited multi-backbone ablations and limited evidence for persona transfer/generalization; authors explicitly do not target zero-shot persona transfer.

**Reviewer Scores:**

z9Wn (4): likely unchanged since main issues are not updated in manuscript.
6KAu (8): likely unchanged
PbEY (6): likely unchanged
b3zB (6): likely unchanged

---

### Decision · Program_Chairs · 2026-01-26

Accept (Poster)